# Drug-induced chromatin accessibility changes associate with sensitivity to liver tumor promotion

Antonio Vitobello[1,3,]*, Juliane Perner[1,]*, Johanna Beil[1], Jiang Zhu[2], Alberto Del Río-Espínola[1], Laurent Morawiec[1], Magdalena Westphal[1], Valérie Dubost[1], Marc Altorfer[1], Ulrike Naumann[1], Arne Mueller[1], Karen Kapur[1], Mark Borowsky[2], Colin Henderson[4,6], C Roland Wolf[4,6], Michael Schwarz[5,6], Jonathan Moggs[1,6], Rémi Terranova[1]

Liver cancer susceptibility varies amongst humans and between experimental animal models because of multiple genetic and epigenetic factors. The molecular characterization of such susceptibilities has the potential to enhance cancer risk assessment of xenobiotic exposures and disease prevention strategies. Here, using DNase I hypersensitivity mapping coupled with transcriptomic profiling, we investigate perturbations in *cis*-acting gene regulatory elements associated with the early stages of phenobarbital (PB)-mediated liver tumor promotion in susceptible versus resistant mouse strains (B6C3F1 versus C57BL/6J). Integrated computational analyses of strain-selective changes in liver chromatin accessibility underlying PB response reveal differential epigenetic regulation of molecular pathways associated with PB-mediated tumor promotion, including Wnt/β-catenin signaling. Complementary transcription factor motif analyses reveal mouse strain–selective gene regulatory networks and a novel role for Stat, Smad, and Fox transcription factors in the early stages of PB-mediated tumor promotion. Mapping perturbations in *cis*-acting gene regulatory elements provides novel insights into the molecular basis for susceptibility to xenobiotic-induced rodent liver tumor promotion and has the potential to enhance mechanism-based cancer risk assessments of xenobiotic exposures.

## Introduction

Hepatocellular carcinoma (HCC) is the most common primary malignancy of the liver. Chronic liver disease due to hepatitis B virus or hepatitis C virus and alcohol accounts for the most HCC cases, but HCC has a multitude of etiological risk factors, including the baseline genetic and epigenetic makeup and multiple environmental cues such as aflatoxin and exposure to chemicals and pharmaceuticals (Gariboldi et al, 1993; Dragani, 2010; Ghouri et al, 2017). Hepatocarcinogenesis is an extremely complex, multistep process involving prominent environment-induced genetic and epigenetic alterations ultimately leading to malignant transformation of the hepatocytes (Farber, 1984; Pogribny & Rusyn, 2014). Although key pathways associated with HCC initiation are emerging, the molecular basis for strain-, gender- or species susceptibilities to HCC is still poorly understood. The identification of the underlying mechanism-based molecular alterations that drive hepatocyte transformation and promote development and progression of HCC is critical for its detection, therapeutic intervention, prevention, and for cancer risk assessment of chemical and pharmaceutical products.

Epigenetic variation, together with genetic variation and metagenomic variation, represent key drivers of phenotypic variation in health and disease. Notably, epigenetic signatures are strongly influenced by the environment and determine phenotypic responses in health and disease (Jirtle & Skinner, 2007; Feil & Fraga, 2012). Epigenetic alterations can occur during the early stages of malignancy, plausibly remaining latent until further stimulation by endogenous or environmental factors (Feinberg et al, 2016; Vicente-Duenas et al, 2018). Multiple layers of epigenetic marks and mechanisms control genome function (Allis & Jenuwein, 2016). Methods for profiling chromatin accessibility, including DNase I hypersensitivity mapping, enable the identification of *cis*-acting gene regulatory elements (or cistrome) that are important for determining cell type identity and functions. The cistrome includes well-characterized DNA sequence elements such as enhancers and promoters and is strongly enriched for transcription factor (TF)-binding sites, providing critical genome regulatory information, but not limited to gene expression regulation (Consortium, 2012; Shen et al, 2012; Thurman et al, 2012; Johanson et al, 2018). Notably, recent integration of regulatory DNA and disease- and trait-associated genetic variant catalogs have demonstrated a disproportionate (>80%) enrichment of disease-associated genetic variants in noncoding enhancer regions. The genetic variants were reported to disrupt

[1]Novartis Institutes for BioMedical Research (NIBR), Basel, Switzerland  [2]NIBR, Cambridge, MA, USA  [3]Inserm, Unité Mixte de Recherche (UMR) 1231, Université de Bourgogne-Franche Comté, Dijon, France  [4]School of Medicine, Jacqui Wood Cancer Centre, Ninewells Hospital and Medical School, University of Dundee, Dundee, UK  [5]Department of Toxicology, University of Tübingen, Tübingen, Germany  [6]Innovative Medicines Initiative MARCAR Consortium (http://www.imi-marcar.eu/index.php)

Correspondence: remi.terranova@novartis.com
*Antonio Vitobello and Juliane Perner contributed equally to this work

important cell type–specific TF regulatory interactions (Maurano et al, 2012; Hnisz et al, 2013; Pennacchio et al, 2013; Mifsud et al, 2015) and were associated with a broad range of phenotypic effects, sometimes driven by subtle effects on target gene expression (Corradin et al, 2014; Soldner et al, 2016). The perturbation of regulatory regions, including enhancers, plays an important role during the tumorigenic process (Davie et al, 2015; Sur & Taipale, 2016). Thus, mapping such perturbations during spontaneously occurring or environmentally driven carcinogenesis has high potential for identifying early mechanism-based biomarkers of carcinogenesis (Thomson et al, 2014; Luisier et al, 2014b).

Several rodent models have been used in defining the pathogenesis of HCC and have contributed to the current knowledge of HCC (Heindryckx et al, 2009; Santos et al, 2017). Treatment of mice with phenobarbital (PB) represents one of the best characterized models of xenobiotic-induced liver tumor promotion and has been extensively evaluated to investigate the kinetics and molecular drivers associated with drug-induced rodent non-genotoxic hepatocarcinogenesis. We previously reported that PB-mediated liver tumor promotion is accompanied by significant progressive transcriptional, epigenetic (DNA methylome and hydroxymethylome), and TF regulatory changes preceding liver-specific tumorigenic events, some of the changes plausibly reflecting dedifferentiation/reprogramming of hepatocytes towards a stem cell–like state (Bachman et al, 2006; Phillips et al, 2009a; Lempiainen et al, 2011, 2013; Thomson et al, 2012, 2016; Luisier et al, 2014b).

Although commonly considered a "rodent" non-genotoxic liver carcinogen, significant species and strain differences (qualitative and quantitative) have been observed for PB-mediated liver tumor promotion, highlighting preexisting differences in baseline tumor susceptibilities (Table S1) (Peraino et al, 1973; Becker, 1982; Diwan et al, 1986; Goldsworthy & Fransson-Steen, 2002). PB positively selects for $\beta$-catenin (Ctnnb1)-mutated liver tumors in the mouse, although inhibiting the outgrowth of mouse liver tumors that harbor an activated MAPK-pathway (i.e., Ha-ras or B-raf–mutated) (Aydinlik et al, 2001; Calvisi et al, 2004). Sensitive mouse strains (e.g., C3H or the hybrid strain B6C3F1) develop Ctnnb1-mutated neoplasms within 12 mo of PB exposure with high incidence, whereas tumor-resistant strains (e.g., C57BL/6) only develop liver tumors after an initiating mutagenic event and long-term PB exposure (Table S1 and references within).

At the molecular level, divergent transcriptional signatures have been identified in mouse strains exhibiting differential sensitivity to PB-driven tumor promotion effects (Phillips et al, 2009b), further supporting the impact of genetic and epigenetic variation on TFs and gene regulatory networks responsible for differential transcriptional readouts in response to PB.

Here, we have exploited established mouse strain–specific differences in sensitivity to PB-mediated tumor promotion to explore drug-induced chromatin regulatory and transcriptional variations underlying phenotypic responses at early stages of liver tumor promotion. We used genome-wide DNase I hypersensitivity profiling (DNase-seq) of liver tissue after treatment of mice with tumor-promoting doses of PB in both tumor-resistant C57BL/6J and tumor-prone hybrid B6C3F1 (C57BL/6 female × C3H/He male) strains. Our analysis reveals quantitative strain differences in hepatic molecular responses to PB at both transcriptional and chromatin

accessibility levels, including genes that regulate Wnt/$\beta$-catenin signaling. Most PB-mediated chromatin accessibility changes occurred at distal intergenic (IG) regions that were on average ~45 kb away from the nearest gene transcriptional start site (TSS), plausibly mapping to the location of cis-acting gene regulatory elements. The analysis of TF motifs underlying these predominantly strain-selective changes in chromatin accessibility highlights several novel candidate transcriptional co-regulators that may underlie the sensitivity of B6C3F1 mice to PB-mediated liver tumor promotion. These data also highlight the significant potential of mapping tissue-specific changes in cis-acting gene regulatory elements for providing novel insights into the molecular basis of xenobiotic-induced phenotypes, including the potential to enhance mechanism-based cancer risk assessments of xenobiotic exposures.

# Results

### Experimental model for comparing mouse strain sensitivity to PB-mediated tumor promotion

Molecular profiling of well-characterized mouse strains that exhibit differences in sensitivity to PB-mediated liver tumor promotion represents an ideal model system for identifying early mechanism-based markers of drug-induced liver tumorigenesis. We previously reported kinetic investigations of early hepatic pathological and transcriptional effects associated with PB treatment (ad libitum access to 0.05% [wt/vol] in drinking water for up to 91 d of treatment) in B6C3F1 (Lempiainen et al, 2013) and C57BL/6J (Luisier et al, 2014a) mouse strains. In these studies, PB concentrations in plasma and liver were determined by liquid chromatography–mass spectrometry and showed comparable and stable plasma levels of PB over time (Fig S1). This treatment regimen reportedly promotes high incidence of tumor formation exclusively in the B6C3F1 mouse strain in absence of mutagenic priming events (Table S1). Despite such differences in tumor promotion effects after long-term (≥1 yr) treatment, the liver histopathology phenotype induced by treatment at earlier time points was essentially identical in both strains and limited to hepatocellular hypertrophy (primarily of perivenous hepatocytes in the central zone of the lobule), starting from 8 d of PB treatment and increasing in severity at later time points (Lempiainen et al, 2013; Luisier et al, 2014a). Thus, this experimental setup enables the investigation of chromatin and transcriptional effects underlying early events of liver tumor promotion sensitivity.

### Differential chromatin accessibility maps of PB-treated mouse livers

We hypothesized that the mapping of PB treatment-mediated hepatic transcriptional and epigenetic effects ($\delta$ = PB response), and their comparison across mouse strains ($\Delta$ = strain-selective PB effects) may enable the characterization of key gene regulatory elements and associated TFs underlying differential tumorigenicity outcomes after chronic PB exposure (Fig 1A).

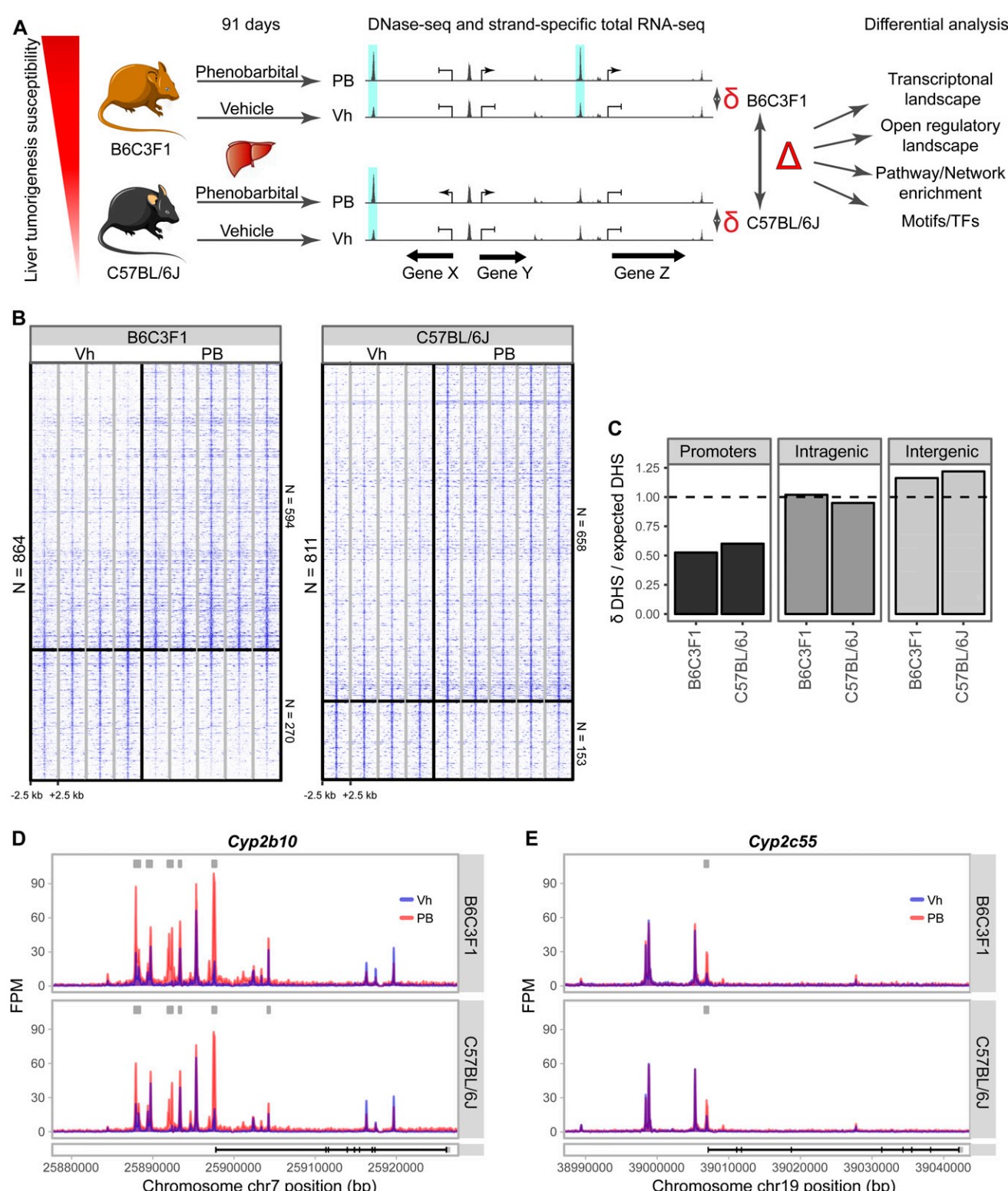

**Figure 1. DNase hypersensitivity mapping of PB-treated mouse (B6C3F1 and C57BL/6J) livers.**
**(A)** Overview of experimental design and analysis plans. Mouse strains of differential tumor promotion sensitivity were PB-treated for 91 d in comparable conditions and with comparable histopathological outcomes. Their livers were processed to profile transcriptome (RNA-seq) and open chromatin landscape (DNase-seq). PB response effects (δ) are compared across strains (Δ) using different computational approaches. **(B)** Profiles of PB-mediated open chromatin changes (|log2 FC| ≥ 0.58, FDR < 0.01) in livers of B6C3F1 and C57BL/6J. Windows of 5 kb (−2.5 to +2.5 kb) centered around each identified δ-DHS are illustrated. Log2-transformed DHS fragment counts in consecutive 50-bp bins within the 5-kb window scaled to average library size are shown. Color intensity from low to high

Mapping chromatin accessibility landscapes can help elucidate key genome regulatory regions (e.g., gene promoters and enhancers) and associated epigenetic mechanisms of gene regulation (Consortium, 2012; Thurman et al, 2012). We, thus, profiled gene expression and chromatin accessibility landscapes in tumor-resistant C57BL/6J and hybrid tumor-prone B6C3F1 male mice after 91 d of PB treatment using archived tissues from our previously reported kinetic studies (Lempiainen et al, 2013; Luisier et al, 2014a).

To gain insight into the gene regulatory landscape accounting for PB-mediated responses, we performed DNase I digestion combined with high-throughput sequencing (DNase-seq) on individual liver samples isolated from control and treatment groups (n = 4–5) of C57BL/6J and B6C3F1 mice. A consensus set of 98,170 DNase I hypersensitive sites (DHSs) was identified that show significant DHS signal in at least three samples. The obtained DNase-seq profiles are overall comparable with 8-wk-old mouse liver DNase-seq publicly available from ENCODE (Pearson's correlation coefficients 0.86–0.91; Fig S2A). DHS signals from individual samples show high degree of reproducibility across samples (Pearson's correlation coefficients 0.55–0.98). The unsupervised clustering and Principal Component Analysis (PCA) based on the top 5,000 most variable DHSs (Fig S3A and B) show that strain background (PC1) is the strongest source of variation among samples, whereas the treatment effects is less strong but consistent (PC3). Overall, this indicates that DNase-seq is a robust method to detect strain- and PB-mediated variations in accessible chromatin regions genome wide.

Using library size–adjusted read counts at each DHS as direct quantitative readout of chromatin accessibility, we performed differential DHS analysis between vehicle- and PB-treated sample groups ($\delta$-DHS). Using |log2 Fold Change (FC)| ≥ 0.58 and False Discovery Rate (FDR) ≤ 0.01 as cutoffs, we found a comparable number of PB-mediated differentially accessible $\delta$-DHSs in B6C3F1 (n = 864) and C57BL/6J (n = 811) and comparable direction of increased/reduced chromatin accessibility effects across strains (Fig 1B and Table S2). Interestingly, we identified $\delta$-DHSs at both promoter and upstream regulatory regions of well-characterized transcriptionally induced PB-responsive and constitutive androstane receptor (CAR)–regulated target genes, such as *Cyp2b10* and *Cyp2c55* (Honkakoski & Negishi, 1997; Konno et al, 2010; Lempiainen et al, 2011) (Fig 1D and E).

To investigate which fraction of the functional genome was most affected by treatment-related changes, we first mapped the $\delta$-DHSs to annotated promoters, intragenic and intergenic regions. We found overall enrichment of the $\delta$-DHSs in intergenic regions and under-representation at promoter regions (here defined as the 1,000-bp region upstream of the Transcriptional Start Site (TSS) (Fig 1C), indicating that chromatin accessibility changed mostly at intergenic regulatory elements. Next, we investigated the activity status of the $\delta$-DHSs at baseline using the histone modification profiles (H3K4me3, H3K4me1, H3K27ac, and H3K9ac) from the 8-wk-old mouse liver in ENCODE, which showed chromatin accessibility profiles consistent with our data. The integration of these four histone modification readouts to the open DHS landscape enables a functional partitioning of the genomic landscape (including constitutively opened promoter regions and tissue-specific active, poised, or silent enhancer regions) (Ram et al, 2011; Shlyueva et al, 2014). Aligning the histone modification profiles to the consensus set of DHSs, thus, revealed expected functional clustering of promoters (cluster I), active (cluster II), poised (cluster III), and inactive enhancers (cluster IV) (Fig S2B). This analysis revealed that the PB-mediated changes in open chromatin landscape in both strains were strongly enriched in cluster IV, whereas depleted at active chromatin clusters I and II (Fig S2C). We, however, noted that functionally active enhancer regions (cluster II) tended to show stronger enrichment for B6C3F1 over C57BL/6J, suggestive of a strain-specific effect in PB-mediated $\delta$-DHS distribution.

Overall, our DNase-seq data robustly identifies a landscape of PB treatment–related regulatory chromatin changes. In both mouse strains, most $\delta$-DHSs indicate increased chromatin accessibility and are enriched in intergenic regions, plausibly representing poised or inactive enhancer regions in the mouse liver.

### Quantitative strain differences in PB-mediated liver chromatin accessibility changes

To evaluate the consistency of global chromatin accessibility effects in both strains, we projected the $\delta$-DHSs (from Fig 1B) over the consensus DHS landscape in both strains (Fig 2A). We found that a subset of $\delta$-DHSs is common to both strains (n = 319). These "shared" effects tend to occur in regions of strongest DHS log2 FC (Fig 2B) and are most consistent across replicates (Fig 2C). A comparable number of $\delta$-DHSs only reached the cutoffs in one of the strains, with 545 and 492 $\delta$-DHSs selectively enriched in B6C3F1 and C57BL/6J, respectively (Fig 2A). Direct visual comparison of the DHS landscapes (Fig 2C) points to coherent trends of the selectively enriched chromatin accessibility changes in both strains. Still, the differential analysis highlights differences of the effects in magnitude and/or consistency across replicates (only reaching the statistical thresholds in one of the strains). Taken together, the observed $\delta$-DHSs are consistent with discrete cell-autonomous differences in molecular regulation and/or changes in small subsets of liver cells, rather than large changes in tissue composition or lineage identity after PB treatment.

Many coding genes are well annotated with their biological functions. Noncoding regions, however, typically lack such annotation. We next associated each $\Delta$-DHS to the nearest TSS (Fig 2D), which is on average 45 kb away and can be located over 1,000 kb away (Fig 2E). Notably, whereas the shared DHSs follow the same bimodal (TSS and distal) distribution as the whole-genome DHS landscape, the strain-selective $\Delta$-DHS landscape tends to distribute away from TSS, possibly accounting for distal enhancer-based changes (Fig 2E). This DHS–TSS proximity calling enabled us to build gene lists for each shared or strain-selective DHS group among which several well-known gene targets of PB signaling were identified (Table S2).

represents signals within the range (1.5, 7). Individual samples are separated by grey lines. **(C)** Relative genomic distribution of $\delta$-DHS with respect to UCSC genome annotation. **(D, E)** Genome browser tracks show $\delta$-DHS effects at the promoter and proximal regulatory regions of two PB-responsive genes *Cyp2b10* and *Cyp2c55*.

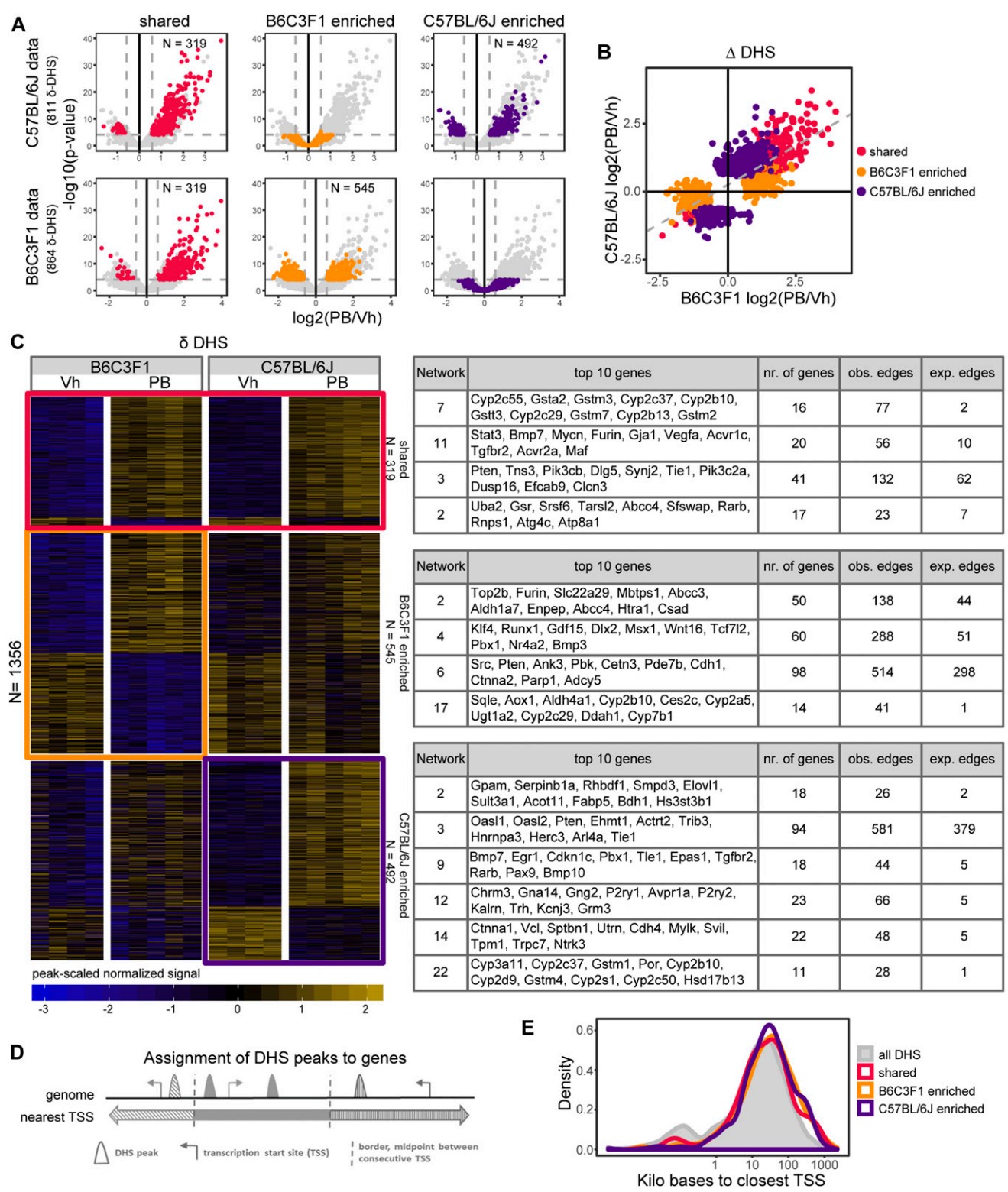

**Figure 2. Quantitative strain-selective chromatin accessibility effects (Δ-DHS) upon PB treatment in mice.**
**(A)** Volcano plot showing the log2 FC and *P*-values of all genes (light grey) from the differential tests using the C57BL/6J samples (first row) or the B6C3F1 samples (second row). The 319 common (red), 545 B6C3F1-enriched (orange) and 492 C57BL/6J-enriched (purple) differentially expressed genes are highlighted. Dashed lines indicate selected statistical cutoffs (|log2 FC| = 0.58 and *P*-value corresponding to FDR = 0.01). **(B)** Linear correlation of Log2 FC PB treatment DHS effects across strains. Only Δ-DHS effects supported by either strain are shown and used to fit the linear regression line indicated by the dashed line. **(C)** Head-to-head comparison of PB-mediated δ-DHS effects, highlighting shared (n = 319, red box) and unique strain effects (n = 545 for B6C3F1, orange box and n = 492 for C57BL/6J, purple box). Δ-DHS effects were

Acknowledging the limitations of DHS–TSS proximity calling and the risk that a fraction of gene hits may not be biologically relevant, we further analyzed the gene hits to identify common underlying biological functions within each shared or strain-selective gene group. We performed network enrichment analysis using the Search Tool for the Retrieval of Interacting Genes/Proteins (STRING database) (Jensen et al, 2009) to identify enrichment in protein–protein interaction partners. Consistent with the well-characterized PB-mediated hepatic xenobiotic response, STRING-db sub-networks enriched in shared DHS–TSS highlighted phase I metabolism genes (e.g., Cyp2b10, Cyp2c55, etc.) as well as signal transduction genes (e.g., Stat3, Pten, etc.) (Fig 2A and Table S3). The most connected genes in the STRING-db sub-networks enriched in B6C3F1 Δ-DHS–associated genes pointed out important regulatory factors, including Klf4, a key regulator of stem cell pluripotency (Takahashi et al, 2007; McConnell & Yang, 2010), Wnt/β-catenin signaling genes such as Wnt16 and Tcf7l2 (alias Tcf4) (network 4), and Src (network 6), which was previously reported to contribute PB-mediated mode-of-action, including through β-catenin signaling regulation (Groll et al, 2016). The C57BL/6J Δ-DHS group showed unique Ctnna1 (network 14) enrichment and other effects related to metabolism, cell cycle, and differentiation (Fig 2A and Table S3).

We previously reported CAR and Wnt/β-catenin signaling–dependent transcriptional effects at the pluripotency-associated Dlk1-Dio3 imprinted gene cluster noncoding RNAs in the liver of mice treated with tumor-promoting doses of PB (Lempiainen et al, 2013; Pouche et al, 2017). Importantly we identify a minimal, but significant B6C3F1-specific increase in chromatin accessibility within the Dlk1-Dio3 cluster upstream of the Meg3 locus and at close proximity (within 500 bp) to the reported imprinting control region (Nowak et al, 2011), consistently we also measured enhanced Meg3 transcriptional up-regulation in B6C3F1 liver material (Table S2 and Fig S4).

### Strain-selective PB-mediated liver transcriptional changes

To enable further functionalization of the DHS open regulatory landscape effects, we ran genome-wide RNA sequencing from matching liver samples. The RNA sequencing data were strongly clustered per strain and treatment effects, and PCA analyses of the top 1,000 most variable genes also show that strain background (PC1) and PB response (PC2) together account for most of the variation in the data (Fig S3C and D). Using cutoffs of log2 FC ≥ 0.58 and FDR < 0.01, we identified 127 differentially expressed genes in C57BL/6J and 269 in B6C3F1 samples (Table S4). Akin to the DHS landscape, we find predominant transcriptional perturbations in a strain-selective manner. 102 genes are commonly regulated and tend to affect genes associated with the strongest transcriptional expression changes, whereas 167 and 25 genes are selectively modulated in B6C3F1 and C57BL/6J, respectively (Fig 3A and B). The effects are qualitatively consistent across strain and replicates (Fig 3C), thus again highlighting that strain differences pertain to quantitative effects in these total liver analyses.

Gene ontology (GO) term over-representation analyses showed enrichment for genes relevant to xenobiotic response in both strains (data not shown), whereas STRING-db protein–protein interaction enrichment analyses highlighted biologically significant B6C3F1-specific Δ-RNA changes. For example, of genes relevant to PB-mediated regulation of Wnt/β-catenin signaling (Cdh1 or Src in networks 4 and 2) and other genes involved in important signaling functions in the liver such as Tgfbr2 (network 4), a member of the canonical Smad-dependent TGF-β signaling cascade, involved in hepatic progenitor cell activation (Ding et al, 2013). As evidenced above by Δ-DHS/associated genes and RNA effects, some notable B6C3F1-specific effects are concordant, including Src, Cdh1, or Tgfbr2, and may represent predominantly strain-enriched signatures of functional relevance (compare network lists in Figs 2C and Figs 3C and see gene loci DHS and RNA effects illustration in Fig 4A and B [Cdh1] and Fig 4D and E [Src]).

### Strain-selective chromatin, transcriptional, and protein expression changes in the β-catenin pathway

By systematically overlapping Δ-RNA and genes with nearby Δ-DHS changes, we characterized 29 of 167 (17% overlap, B6C3F1) and 3 of 25 (12% overlap, C57BL/6J) genes of concurrent RNA and DHS effects (Fig 3D). Because the chromatin at gene promoters is by and large constitutively opened and from the overall distal distribution of δ-DHSs (Figs 1 and S2B and C), a limited functional overlap between DHS/gene expression changes is expected. We also evaluated whether alternative DHS gene assignment approaches may yield different functional overlap output. Scanning the whole-genome TSS (n = 34,219) for δ-DHS association in flanking windows of ±5–100 kbp identified 10–26% functional differential RNA/DHS overlap in B6C3F1 and 4–24% overlap in C57BL/6J (Fig S5A). In a complementary approach, scanning the differential DHS landscape for δ-RNA association in flanking windows of ±5–100 kbp identified 4–13% functional RNA/DHS overlap in B6C3F1 and close to null in C57BL/6J (Fig S5B). The limited overlap in the later method is expected from the larger number of DHSs over gene expression changes in both strains. Thus, overall using a range of gene/DHS assignment approaches yields equivalent overlap, in the range of 10–20%. Interestingly, a large number of DHSs can not immediately be assigned to detectable gene transcriptional changes (Fig 5A) and yet may represent a genomic landscape of functional interest (see below).

Notably, STRING-db and literature analysis of the overlapping gene loci highlight evidence for functional and/or biochemical interactions with Wnt/β-catenin signaling functions (e.g., Cdh1, Klf9, Src, Tgfbr2, Ddah1, and Ank3), a pathway strongly reported to mechanistically contribute to PB-mediated liver tumor promotion (Aydinlik et al, 2001; Rignall et al, 2011; Groll et al, 2016), overall suggesting functional relevance of this concordant Δ-strain landscape. To verify that DHS and RNA changes (Fig 4) represent

---

mapped to the nearest genes and STRING-db protein–protein interaction sub-networks enriched in each group are indicated (full list available from Table S3). **(D)** Illustrative summary of DHS peaks to gene assignment approach. A single gene was associated to each Δ-DHS through associating the nearest TSS. The full list of DHS-associated genes is available from Table S2. **(E)** Density distribution of distances to nearest TSS for genome-wide DHS landscape compared with PB-mediated δ–DHS for each indicated categories.

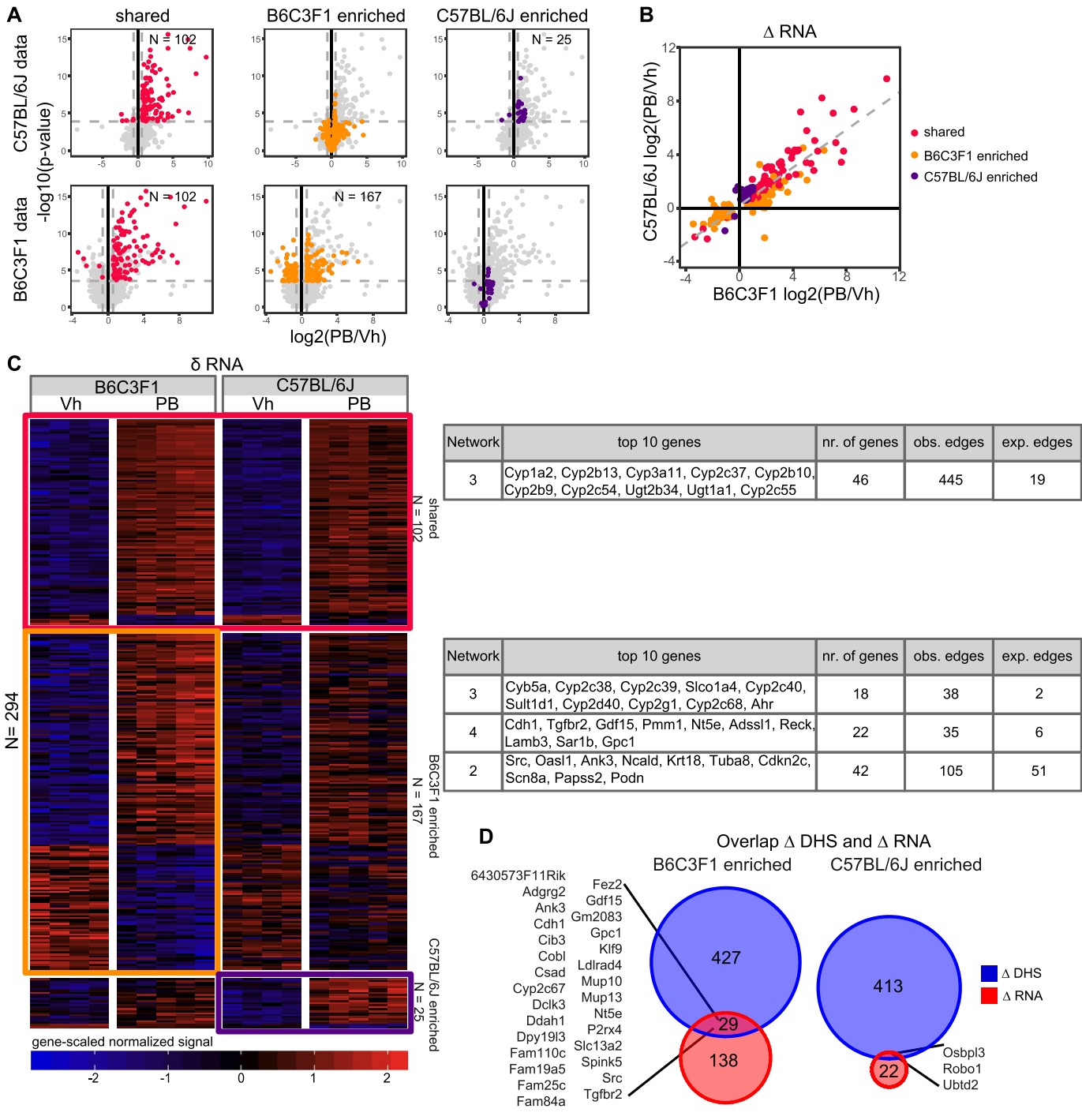

**Figure 3. Strain-selective transcriptional effects (Δ-RNA) upon PB treatment in mice.**
**(A)** Volcano plot showing the log2 FC and P-values of all genes (light grey) from the differential tests using the C57BL/6J samples (first row) or the B6C3F1 samples (second row). The 102 common (red), 167 B6C3F1-enriched (orange), and 25 C57BL/6J-enriched (purple) differentially expressed genes are highlighted. Dashed lines indicate selected statistical cutoffs (|log2 FC| = 0.58 and P-value corresponding to FDR = 0.01). **(B)** Linear correlation of Log2 FC PB treatment RNA effects across strains. Only Δ-RNA effects supported by either strain are shown and used to fit the linear regression line indicated by the dashed line. **(C)** Head-to-head comparison of PB-mediated δ-RNA effects, highlighting shared (n = 102, red box) and unique strain effects (n = 167 for B6C3F1, orange box and n = 25 for C57BL/6J, purple box). The list of transcriptionally modulated genes is available from Table S4, STRING-db protein–protein interaction sub-networks enriched in each group are indicated (full list available from Table S3). **(D)** The overlay of Δ-DHS–associated genes (nearest TSS assignment approach) and Δ-RNA expression changes highlights a short list of loci displaying both changes in chromatin accessibility and transcriptional modulation, which are unique to each strain.

functionally relevant changes, we next evaluated protein expression levels of a subset of candidates relevant to Wnt/β-catenin signaling and for which well-characterized antibodies are available.

We leveraged matching liver tissue samples and ran semi-quantitative Western blot analyses of both Cdh1 and Src, confirming an increase in protein levels in PB-treated B6C3F1 mice (Fig

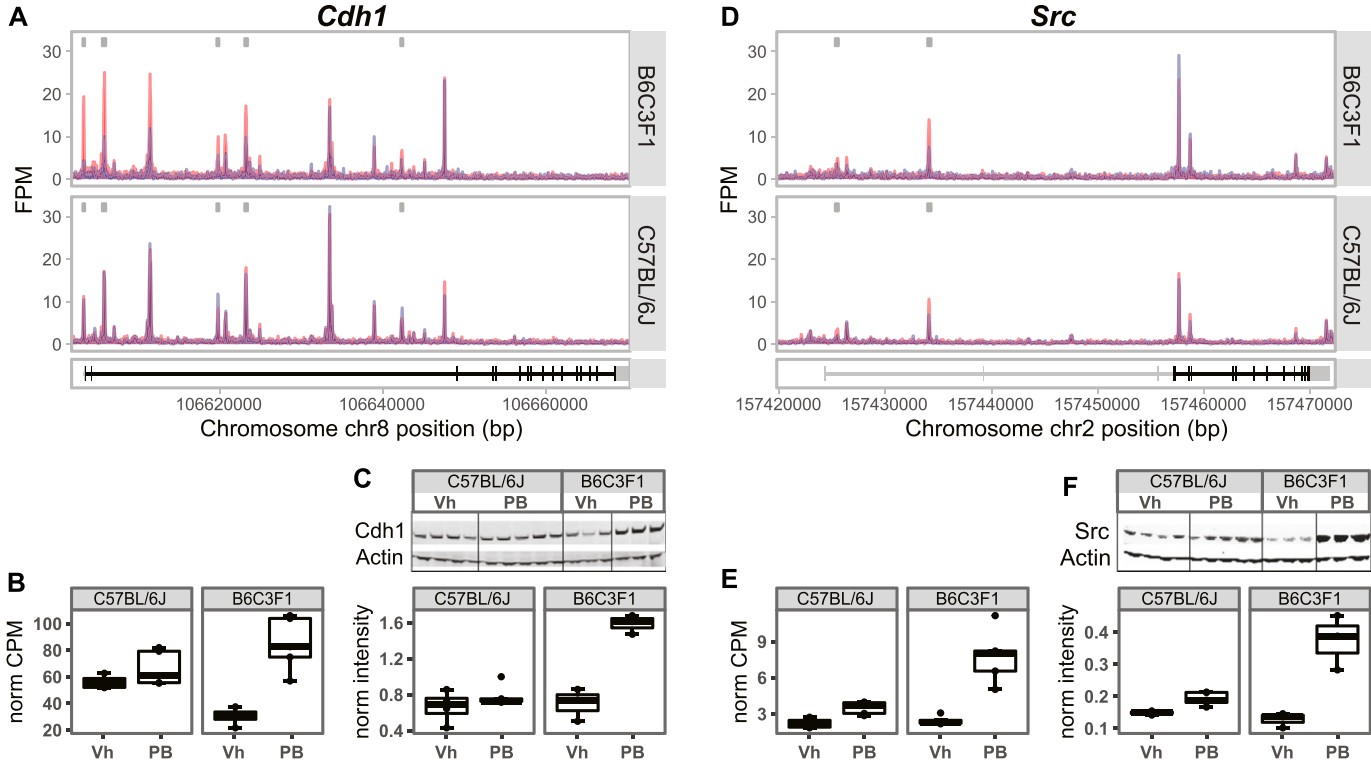

**Figure 4. Strain-selective chromatin and transcriptional effects at Wnt/β-catenin signaling relevant to loci.**
**(A)** Genome browser DHS tracks show B6C3F1-enriched δ-DHS effects at the Cadherin-1 (*Cdh1*) locus. **(B)** RNA-seq, counts per million (CPM) data show B6C3F1-specific increase in *Cdh1* mRNA expression. **(C)** Western blot analysis and quantification show increase in Cdh1 protein expression in liver of PB-treated B6C3F1 animals. **(D)** Genome browser DHS tracks show B6C3F1-enriched Δ-DHS effects at the tyrosine-protein kinase *Src* locus. **(E)** RNA-seq, CPM data show B6C3F1-specific increase in *Src* mRNA expression. **(F)** Western blot analysis and quantification show increase in Src protein expression in the liver of PB-treated B6C3F1 animals. Each light grey box in the genomic browser snapshots represent a δ-DHS peak above statistical cutoffs (Log2 FC ≥ 0.58, FDR < 0.01).

4C and F). These data highlight that a subset of PB-mediated hepatic responses of functional relevance can be discerned at both RNA and chromatin accessibility levels. Chromatin accessibility changes, akin to PB-mediated DNA (hydroxy)methylation effects (Thomson et al, 2013), may thus represent potential early biomarker signatures for non-genotoxic carcinogen exposure.

### Exploring transcriptional regulators of the noncoding, strain-selective Δ-DHS landscape

The vast majority of strain-selective changes in DHSs after PB treatment do not account for proximal changes in gene expression but may still represent important gene regulatory elements underlying mouse strain differences in tumor promotion sensitivity (Fig 5A). The *cis*-acting gene regulatory element landscape is in particular strongly enriched for transcription factor–binding sites (TFBSs) and represents critical genome regulatory information (Consortium, 2012; Shen et al, 2012; Thurman et al, 2012; Johanson et al, 2018). We, therefore, performed TF motif enrichment analysis individually on the up- or down-regulated PB-responsive strain-selective Δ-DHS landscape for each mouse strain. Because TF-binding motifs frequently overlap between different TFs, we used the root motifs from the motif clustering provided in the JASPAR database in our analysis. To strengthen the

robustness of the detected motifs further, we also performed de novo motif analysis and overlapped the results with the motifs detected in the enrichment analysis. The resulting enriched motif clusters with associated similar de novo motifs are shown in Fig 5B and Table S5.

In this analysis, cluster 32 (Stat family members, Bcl6, and Bcl6b) and cluster 19 (Nfat family members and Rbpj), both based on down-regulated B6C3F1 DHS landscape gave the strongest enrichment (low *P*-values, q-values, high FC enrichments, and several good de novo motif predictions). Two additional clusters were identified albeit less robustly using up-regulated B6C3F1-unique DHSs: cluster 11 (Smad family members, Tbx family members, and Srebf1/2) and cluster 12 (Fox family members). Only one cluster was weakly identified from C57BL/6J up-regulated DHS landscape (includes Jun/Fos, Bach, and other TF motifs), despite a similar count of strain-specific open chromatin changes (Fig 2A).

To evaluate the biological function of the different TF motifs clusters, we first predicted gene targets of Δ-strain DHS landscape for each of the enriched TF clusters and ran GO term enrichment analysis (Fig 5C and Table S5). Overall, this analysis highlighted a number of biological functions of potential relevance to early tumor development that were predominantly observed in the B6C3F1 strain after treatment with PB (e.g., change in nuclear functions, metabolic changes, epithelial to mesenchymal transition, cell–matrix adhesion/organization, etc.).

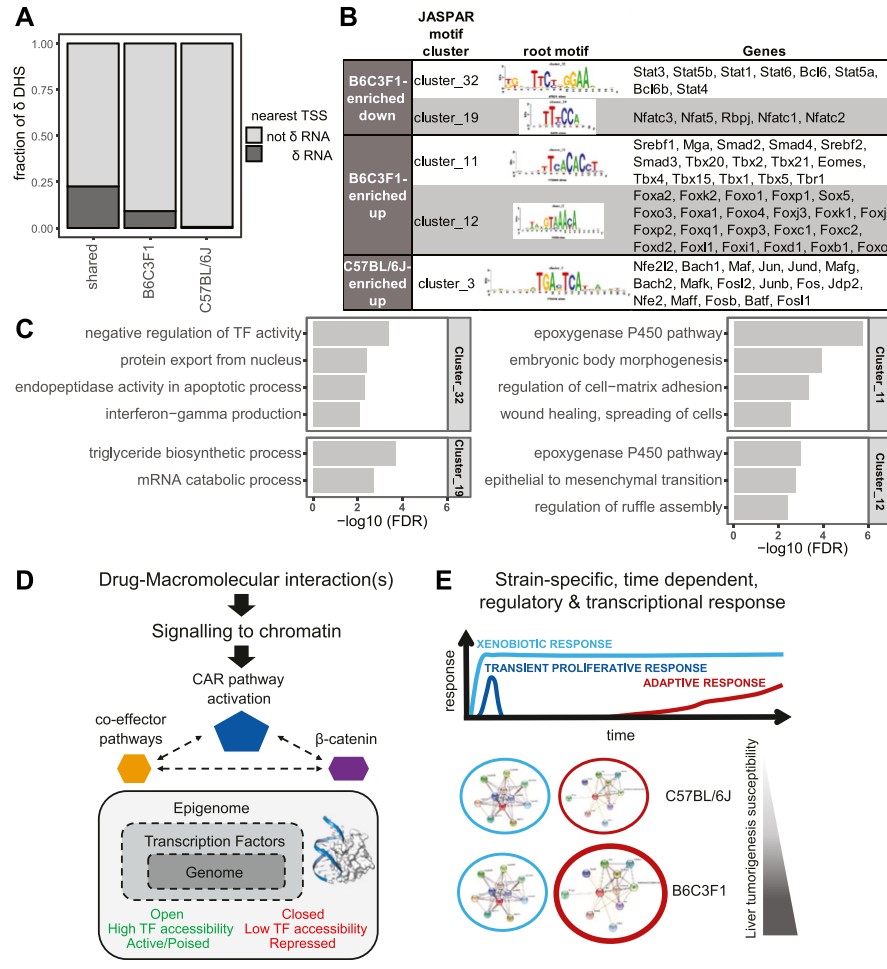

**Figure 5. Strain-selective TF regulatory pathways effects upon PB treatment.**
**(A)** Most Δ-DHSs are distal to gene TSS and not associated with significant changes in the nearest gene expression (δ-RNA). **(B)** Summary of TF motif enrichment in down-regulated and up-regulated Δ-DHSs relative to the full consensus set of detected DHSs. Four clusters of motifs are shown for B6C3F1 and one for C57BL/6J (the full list available from Table S5). The genes shown are expression-filtered members of the motif cluster sorted by average expression. **(C)** GO biological pathway analysis of predicted TF target sites in the respective Δ-DHS group. **(D)** Updated model from Lempiainen et al (2013), Pouche et al (2017) highlighting the importance of the signaling to chromatin and novel potential TF regulatory pathways (baseline and treatment related) in response to CAR-activating xenobiotic exposure. **(E)** Predominantly strain-selective β-catenin and additional co-effector regulatory pathway effects (red) are likely contributing to the chronic and long-lasting adaptive response associated to increased tumor promotion sensitivity in B6C3F1 mice.

To help filter and assess functional relevance of the identified TF motifs, we next examined individual TFs using different knowledge-based sources such as STRING-db, GeneCards database, and literature assessments. We systematically interrogated the prior knowledge towards relevance of the TFs in PB response, CAR activation, Wnt signaling, and/or hepatocarcinoma.

Stat proteins are a family of seven TFs that form part of the JAK-STAT signaling cascade. Significant evidence support functional implication of Stat family members in hepatocarcinoma and cross-interactions with some of the core genes and pathways identified from PB-mediated changes in RNA or open chromatin effects (such as Src and TGF-β), ultimately consistent with the early tumor promotion relevance of this cluster of TF motifs in B6C3F1 mice (see the Discussion section). Notably, Bcl6, in the same cluster 32, is strongly reported to functionally interact with Stat family members, including through the regulation of interferon-γ (Madapura et al, 2017), representing one of the enriched GO biological pathway for this cluster (Fig 5C and Table S5). Consistent with its decreased motif activity, the tumor suppressor Bcl6B was also previously identified to inhibit HCC metastases in vitro and in vivo (Wang et al, 2014). Likewise, and also consistent with decreased motif activity, NFAT family members such as NFTAc1 are reported as tumor suppressor in HCC, through inducing tumor cell apoptosis (Xu et al,

2018). Examining the up-regulated DHS-associated TF motifs (clusters 11 and 12) also highlights core functional and biochemical associations in liver cell proliferation control and HCC of TGFβ/Smad proteins and pathway (Yoshida et al, 2014) or of selected Fox TF members. For example, Foxo1 is known to be under the regulatory control of CAR (Kazantseva et al, 2014) and a direct interactor of β-catenin in certain conditions (Liu et al, 2011).

Overall, these analyses highlight TF families that may play a role in determining mouse strain–specific liver tumorigenic responses to PB (Fig 5D and E). Despite an overall equivalent number of Δ-DHS signatures in both strains, the TF motifs are particularly enriched in PB-treated B6C3F1. Notably, some of these TFs have previously been associated with liver tumor promotion (including the Wnt/β-catenin signaling pathway), whereas others represent potentially novel regulatory factors for this hepatic phenotype.

# Discussion

Gene regulation depends on the coordinated interplay of local and long-range chromatin interactions within the three-dimensional space of the nucleus. In particular, it has been noted that

noncoding regions of the genome contain *cis*-acting gene regulatory elements (e.g., enhancers) that determine cell type–specific functions through interactions with TFs. Importantly, the perturbation of enhancers has been associated with tumor development (Davie et al, 2015; Sur & Taipale, 2016). Enhancers are characterized by cell type–specific open chromatin structures that enable access for specific TFs and can be efficiently mapped using nuclease-based assays coupled with deep sequencing. In this study, we have exploited well-characterized mouse strain–specific differences in sensitivity to PB-mediated liver tumor promotion (Table S1) to investigate the underlying molecular mechanisms that could account for differences in phenotypic outcome. We have focused on liver tissue–based genome-wide profiling of chromatin accessibility (DNase-seq) and gene expression (RNA-seq) to characterize early molecular perturbations of *cis*-acting gene regulatory elements after exposure to tumor-promoting doses of PB. We identify quantitative differences in PB-induced hepatic molecular pathways and gene regulatory networks between two mouse strains that may account for differences in their sensitivity to liver tumor promotion. These observations also provide an entry point for investigating potential links between mouse strain–specific genetic variation and liver-specific gene regulatory landscapes after PB exposure.

Our molecular analysis on total liver reveals PB-induced chromatin accessibility and transcriptional changes that are either common to resistant (C57BL/6J) versus sensitive (B6C3F1) mouse strains or predominantly strain selective. Most chromatin accessibility changes take place at distal regulatory regions consistent with an important role of *cis*-acting gene regulatory elements in mediating early hepatic molecular responses to PB. We interrogated the function of the distal regulatory regions by mapping to the most proximal gene annotation (based on the position of the gene TSS) and by investigating TF motifs associated with changes in chromatin accessibility. These complementary approaches were integrated with gene expression and GO-based pathway analyses to highlight candidate groups of genes and/or molecular pathways associated with the early stage of PB-mediated non-genotoxic hepatocarcinogenesis. Importantly, these analyses highlighted quantitative differences between sensitive and resistant mouse strains in molecular pathways known to be associated with PB-mediated liver tumor promotion including Wnt/β-catenin signaling.

In addition, we also identified candidate TF regulators via PB-mediated changes in chromatin accessibility that may represent novel early biomarkers signatures of PB-mediated hepatocarcinogenesis. Our observation of down-regulation of DHSs containing Stat/Bcl6 family TF motifs (cluster 32) is suggestive of reduced Stat signaling activity in PB-treated B6C3F1 mice. There is significant evidence for a role of RAS and JAK/STAT pathway activation in HCC and therapeutic modulation of these pathways for the treatment of human liver cancer is being actively explored (Calvisi et al, 2006). Recent studies have also elucidated specific functions for JAK and STAT protein family members in Non Alcoholic Steato Hepatitis (NASH) and HCC (Grohmann et al, 2018; Kaltenecker et al, 2018). Genetic disruption of Stat5 activity in a mouse model for liver fibrosis was associated with elevated TGF-β levels and enhanced growth hormone-induced Stat3 activity, ultimately

contributing to the development of chronic liver disease (Hosui et al, 2009). Furthermore, STAT3 has been reported to confer hepatoprotective or oncogenic functions depending on the extent and duration of additional stressors, including inflammation (Wang et al, 2011). Consistent with our observations of reduced Stat pathway–associated TF activity in PB-treated mice, Stat5b expression levels were reported to be suppressed in the livers of mice treated with CAR activators, including PB (Oshida et al, 2016). Interestingly, we also observed up-regulation of DHSs having TF motifs (cluster 11) that included Smad family TFs in PB-treated B6C3F1 mice, implying a potential increase in Smad activity. Together, our data would be consistent with a previous report of TGF-β induced SMAD signaling in hepatoma cells (Buenemann et al, 2001), and the potential for upstream regulation of TGF-β by Stat5 (Hosui et al, 2009).

The down-regulation of DHSs containing Bcl6 and Bcl6b TF motifs (cluster 32) implies a reduction in Bcl6-mediated signaling in PB-mediated B6C3F1 mice and is consistent with the low levels of BCL6B expression that have been previously reported as a potential prognostic biomarker for HCC (Wang et al, 2015b). Interestingly, we previously reported Bcl6 as a candidate target of Zfp161 TF motif, whose hepatic activity was computationally inferred to be decreased after PB treatment of mice and hypothesized to participate in the regulation of quiescent hepatocyte G0–G1 transitions during both early and late stages of PB-mediated tumorigenesis (Luisier et al, 2014b).

The identification of TF motifs associated with the Fox family TFs (cluster 12) in up-regulated DHSs is suggestive of increased Fox-mediated signaling in PB-treated B6C3F1 mice. The Forkhead box (FOX) proteins are a family of TFs that respond to a wide range of external stimuli and regulate diverse biological processes both during development and throughout adult life (Golson & Kaestner, 2016). Aberrant activity of Fox family members Foxa1 and Foxa2 was previously reported as central component of sexual dimorphism of mouse HCC (Li et al, 2012). Likewise, the FOXQ1/NDRG1 axis was shown to exacerbate HCC initiation via enhancing cross talk between fibroblasts and tumor cells (Luo et al, 2018), and FOXO1 was reported to contribute to HCC through a different mode of action (Dong et al, 2017; Jia et al, 2018). Notably, FoxO1 was reported to directly cross talk with CAR to regulate p21 expression and cell proliferation (Kazantseva et al, 2014). Foxp1 was also reported as a candidate hepatic target for the Zfp161 TF motif whose hepatic activity was computationally inferred to be decreased after PB treatment of mice (Luisier et al, 2014b).

In addition to highlighting novel TF motifs and pathways, we also identified minimal but significant changes in hepatic chromatin accessibility in PB-treated B6C3F1 mice within a gene regulatory element (an IG differentially methylated region [IG-DMR]) that regulates the *Dlk1-Dio3* imprinted gene cluster. PB was previously reported to progressively up-regulate the expression of noncoding RNAs (including *Meg3*) encoded by the *Dlk1-Dio3* locus in a CAR and β-catenin–dependent manner (Lempiainen et al, 2013). PB-mediated up-regulation of *Meg3* localises to perivenous hepatocytes that have been associated with Wnt-signaling dependent stem cell–like properties (Wang et al, 2015a). *Meg3* expression has also been associated with mouse stem cell pluripotency (Liu et al, 2010; Stadtfeld et al, 2010) and a subset of human HCCs (Luk et al, 2011),

suggesting that CAR activators such as PB may drive dedifferentiation of adult hepatocytes towards a stem cell–like state during the early stages of hepatocarcinogenesis. Interestingly, a hepatocarcinogenesis sensitivity locus *Hcs3* that was originally identified using mouse backcrosses and linkage analysis (Dragani et al, 1991; Gariboldi et al, 1993) maps within a few megabases of the *Dlk1-Dio3* cluster (Drinkwater & Ginsler, 1986; Lempiainen et al, 2013). Our data suggest a model in which PB-mediated transcriptional responses from this epigenetically imprinted gene cluster may be enhanced via increased chromatin accessibility that is mediated by B6C3F1 strain-selective genetic factors.

The potential contributions of PB-mediated changes in hepatic *cis*-acting gene regulatory elements and TFs to tumor promotion in B6C3F1 mice will require further biochemical and functional investigations. Nevertheless, our study highlights several candidate TFs that may act as co-effectors during the early stages of PB-mediated tumor promotion effects in susceptible mouse strains (Fig 5D). Most chromatin accessibility changes take place distally from gene promoters at putative enhancers. Enhancers are bound by regulatory TFs and serve as integrators of intracellular and extracellular signaling pathways to generate cell type–specific transcriptional responses (Shlyueva et al, 2014). The functional binding of regulatory TFs at *cis*-acting gene regulatory elements such as enhancers orchestrates long-range gene regulatory interactions within the three-dimensional space of the nucleus, enabling cell type–specific and spatiotemporal control of gene expression patterns which drive cell identity and function (Thurman et al, 2012; Jin et al, 2013; Zhang et al, 2013). Importantly, distally bound TFs can also regulate genome functions underlying cell identity through mechanisms that are independent of transcription (Johanson et al, 2018). Thus, the characterization of xenobiotic-induced perturbations of accessible chromatin landscapes and associated TF networks holds great potential for the identification of pathophysiologic mechanisms.

We recognize that the characterization of xenobiotic-mediated tissue-specific molecular responses within tissue derived from in vivo model systems is associated with some significant limitations. In particular, determining the specific cell types associated with molecular responses can be challenging. Multiple biological pathways can dynamically respond within and across cell types, and thus, a precise linkage of molecular effects to cellular phenotypes requires follow-up investigations (e.g., single-cell resolution molecular profiling coupled with pharmacologic and/ or genetic modulation of pathway components). The functional annotation of long-range gene regulatory interactions (e.g., enhancers, transcriptional factor interactions, and nuclear organization) will require follow-up locus-/pathway-specific tools and functional analyses, including the direct biochemical mapping of inferred TF occupancy by chromatin immunoprecipitation. Likewise, future work will be necessary to investigate local and global genetic variations in context of strain-selective differences in chromatin accessibility and TF regulation.

The epigenomic landscape is strongly responsive to environmental conditions, including exposure to xenobiotics such as chemicals and pharmaceuticals (Jirtle & Skinner, 2007; Feil & Fraga, 2012) and, thus, represents an important molecular space for

gaining potential insight into pharmacologic and toxicologic responses to xenobiotics (Szyf, 2007; Lewis et al, 2017; Israel et al, 2018). This study highlights the functional and toxicological relevance of hepatic epigenomic responses in a well-characterized rodent model for drug-induced non-genotoxic carcinogenesis. The mapping of mouse strain–selective pertubations of *cis*-acting regulatory elements (cis-trome) provides novel TF–based insights into the molecular basis for susceptibility to PB-induced rodent liver tumor promotion (Fig 5E). These observations are also consistent with recently reported links between genetic variation and tissue-specific epigenetic perturbations after toxicant exposures (Lewis et al, 2017; Israel et al, 2018). Thus, the integration of genetic, epigenomic, and transcriptomic data together with phenotypic endpoints has the potential to enhance mechanism-based safety assessment of both chemicals and pharmaceuticals. In addition, integrated epigenomic and transcriptomic profiling of tissue-specific responses to xenobiotics may also provide a valuable resource for exploring gene regulatory networks underlying broader pathophysiologic processes.

# Materials and Methods

## Ethics statement

In vivo rodent studies were performed in conformity with the Swiss Animal Welfare Law (specifically under the animal licenses No. 2345 by Kantonales Veterinaramt Basel-Stadt [Cantonal Veterinary Office, Basel]).

## Animal treatment and sample preparation

As previously reported in (Lempiainen et al, 2013; Luisier et al, 2014a), male 9–11-wk-old C57BL/6J mice were obtained from Taconic. Animals were allowed to acclimatize for at least 5 d before being randomly divided into two treatment groups of five animals each. 0.05% (wt/vol) PB (Hänseler AG) was administered to one group through ad libitum access to drinking water for either 1, 7, 14, 28 or 91 d. Male B6C3F1/Ctrl (C57BL/6 ♀ × C3H/He ♂) mice 4–5 wk old were obtained from Charles River Laboratories. Animals were allowed to acclimatize for at least 5 d before being randomly divided into two treatment groups of five animals each. 0.05% (wt/vol) PB (Sigma-Aldrich) was administered to one group through ad libitum access to drinking water for 91 d. Individual mice from both studies were checked daily for activity and behavior and sacrificed on the last day of dosing. Livers were removed before freezing in liquid nitrogen and –80°C storage.

## Nuclei preparation, DNase treatment, and DNA purification

DNase I assay was performed as previously described by Ling and Waxman (Ling and Waxman 2013a, 2013b) with minor modifications. 100–150 mg of frozen liver tissue was thawed on ice for 5 min and then homogenized in 1 ml ice-cold nuclear homogenization buffer (NEHB): 10 mM Hepes-KOH, pH 8.0, 25 mM KCl, 1 mM EDTA, 2 M sucrose, 10% glycerol, 0.15 mM spermine, 0.5 mM spermidine, 10 mM NaF, 1 mM orthovanadate, 1 mM PMSF, 0.5 mM DTT, and 1× protease

inhibitor cocktail; using a 7-ml Dounce tissue homogenizer. Nuclei were isolated by ultracentrifugation in an SW 40 Ti rotor, overlying the homogenized tissue on top of 5 ml NEHB in a 14-ml Thinwall Ultra-Clear 14 × 95-mm tube (cat. no. 344060; Beckman Coulter). Centrifugation was performed at 66,000$g$ for 1 h at 4°C. Isolated nuclei were rinsed three times in 1 ml ice-cold buffer A: 15 mM Tris–HCl, pH 8.0, 15 mM NaCl, 60 mM KCl, 1 mM EDTA, 0.5 mM EGTA, 0.5 mM spermidine, and 0.3 mM spermine. The pelleted nuclei were suspended in digestion buffer (nine volumes of buffer A and one volume of 10X DNase I digestion buffer [60 mM $CaCl_2$, 750 mM NaCl]) by using a Dounce homogenizer, counted using a hema-cytometer after 100-fold dilution in PBS, and then resuspended in final aliquots of 250 $\mu$l containing ~3.3 × $10^6$ nuclei each. DNase I digestion was carried out at 37°C for 3 min using a final concentration of 50 U/ml of RQ1 RNase-free DNase I (cat. no. M6101; Promega). We digested $10^7$ nuclei per mouse. DNase digestion was stopped using 300 $\mu$l of stop buffer: 50 mM Tris–HCl, pH 8.0, 100 mM NaCl, 0.1% SDS, 100 mM EDTA, 0.5 mM spermidine, and 0.3 mM spermine. DNA purification was performed at 55°C overnight using 10 $\mu$l of 10 mg/ml proteinase K. RNA digestion was carried out at 37°C for 30 min using 5 $\mu$l of 10 mg/ml RNase A. DNA from different aliquots deriving from the same animal were pooled and subsequently extracted using an equal volume of phenol–chloroform–isoamyl alcohol and then one volume of chloroform–isoamyl alcohol. The solution recovered from the phe-nolic extraction was adjusted with 5 M NaCl to give a final concen-tration of 0.8 M NaCl and then DNA fractionation was performed in a SW 40 Ti rotor, overlying the adjusted solution on top of 9 ml of a sucrose step gradient (40%, 35%, 30%, 25%, 20%, 17.5%, 15%, 12.5%, and 10% solutions in 1× centrifugation buffer: 20 mM Tris–HCl, pH 8.0, 5 mM EDTA, and 1 mM NaCl). Centrifugation was performed at 78,000$g$ in a 14 ml Thinwall Ultra-Clear 14 × 95 mm tube for 24 h at 20°C setting acceleration and deceleration parameters at five. After ultracentrifu-gation, the first 3-ml fraction was recovered and purified over a MinElute Purification Column (QIAGEN) and eluted in 13 $\mu$l of provided Elution Buffer.

## DNase I library preparation and sequencing

DNase I digests the entire genome with a strong preference for regions that are devoid of nucleosomes and unoccupied by TF. Sub-nucleosomal fragments (<145 bp) are enriched for nucleosome-free and TF bound regions, which can be further analyzed to obtain important insights into the functional genomic landscape (Vierstra et al, 2014). We, therefore, size-selected the low molecular fraction of the DNase-seq libraries and performed 76-bp paired-end (PE) sequencing, obtaining an average of 43 million fragments per sample (Table S6). Specifically, next gen-eration sequencing libraries were prepared with the NuGEN Ovation Ultralow System V2 with A-tailing (TECAN) from 3 to 20 ng of input material. Inputs smaller or higher than 10 ng were am-plified using 16 or 13 rounds of PCR, respectively. Up to nine li-braries were pooled before size selection of the 40–250 bp fraction on a BluePippin BDF3010 3% DF Marker Q2 cassette or PippinHT BEF2010 2% DF Marker M1 cassette (both Sage Science) and PE 76-bp sequencing was performed on the HiSeq 2500 using v4 re-agents (Illumina).

## Genome imputation for B6C3F1

To mitigate the impact of strain differences in read mapping and assess chromatin variation across inbred mice (Hosseini et al, 2013), we generated strain-specific reference sequences using known Single Nucleotide Polymorphisms (SNPs) from the Mouse Genomes Project (Keane et al, 2011; Doran et al, 2016). We used the mm10 reference genome (C57BL/6J strain) as reference for the C57BL/6J samples, subselecting chromosomes chr1-19, X and Y using SAMtools faidx v0.1.19 (Li et al, 2009) for analysis. To impute the genome of B6C3F1, we downloaded the SNPs of the parental strain C3H/HeJ from the Mouse Genomes Project (C3H_HeJ.mgp.v5.snps.dbSNP142.vcf). After sorting using VCFtools v0.1.14 (Danecek et al, 2011), we in-troduced the C3H/HeJ-specific SNPs into the mm10 genome using the FastaAlternateReferenceMaker from the Genome Analysis Toolkit (McKenna et al, 2010). We generated index files with SAMtools faidx. Finally, to confirm the expected genotypes for each strain, we compared chromosomal sizes from mm10, C57BL/6J, and C3H/HeJ and manually inspected several loci with the Integrative Genomics Viewer v2.3 (Robinson et al, 2011; Thorvaldsdottir et al, 2013).

## DNase-seq data pre-processing

Libraries were demultiplexed using bcl2fastq version 2.17. Illumina adaptor sequences at the end of the reads were removed using Cutadapt v1.8 and resulting reads with a minimum length of 30 bp were retained. All samples were aligned to the mm10 reference genome using Bowtie2 v2.1.0 (Langmead & Salzberg, 2012) with the following parameters: -I 50 -X 400 –fr –no-mixed –no-discordant. Binary Alignment Map (Bam) files were created from sam files and coordinate sorted using SAMtools. PCR and optical sequencing du-plicates were flagged using Picard Tools' MarkDuplicates v1.107(1667). Only properly paired, unique reads with a map quality score above 12 were retained after filtering using SAMtools. Bam files of the same biological replicate from multiple sequencing runs were merged and again filtered for PCR or optical sequencing duplicates.

In addition, we mapped the B6C3F1 samples to the imputed C3H/HeJ genome and compared the results of the alignments with the two different reference genomes using bamUtil diff with options –mapQual–mate. Under these conditions, no differences were detected in common read pairs, and less than 50–100 read pairs resulted to be unique for one or the other alignment. Hence, we decided to use the alignment to the mm10 reference for all downstream analysis.

## Re-mapping of the encyclopedia of DNA elements (ENCODE) liver data

The mouse DNase-seq and ChIP-seq data were integrated from mouse ENCODE (GSE37074 and GSE31039, respectively). FASTQ files were retrieved from NCBI SRA (https://www.ncbi.nlm.nih.gov/sra) and processed with the same pipeline above.

## DHS calling and differential analysis on DNase-seq data

DHSs were identified and *bedGraphs* were created applying a q-value cutoff (−$q$) of 0.01 using MACS2 v2.1.1.20160309 (Zhang et al,

2008). Differential peak analysis was performed in R 3.2.3 using the Bioconductor package *DiffBind* v1.16.3 (Ross-Innes et al, 2012). Consensus peaks for each condition were created using *dba.counts* requiring that at least three samples show a significant peak in the same genomic window (minOverlap = 3). The *DBA_EDGER* method was used to call differential peaks including treatment and strain in the contrast. Regions showing an absolute log2 fold-change |log2 FC| ≥ 0.58 between selected contrasts and a FDR ≤ 0.01 (Benjamini–Hochberg) are called differential.

### DHS coverage profile

Coverage profiles were generated from the bam files by counting reads in 50-bp bins within the 5-kb region centered at the midpoint of each consensus DHS. The bamProfile function in the bamsignals (v1.12.1) R package was used setting the PE flag to "ignore." The profiles were scaled to the average across samples of the total number of reads falling into the consensus DHSs.

### Overlapping DHSs with gene annotation

To assign DHSs to promoters, intra-, or intergenic elements, the function assignChromosomeRegion in the ChIPpeakAnno v3.16.1 R package (Zhu et al, 2010; Zhu, 2013) was used on the UCSC known gene annotation (https://genome.ucsc.edu/). DHSs assigned to the categories *fiveUTRs*, *threeUTRs*, *Exons*, or *Introns* were called intragenic, whereas DHSs assigned to the categories *Intergenic. Region* and *Immediate Downstream* were called Intergenic (categories as per R package).

### Browser track visualization of DNase-seq data

For visualization of individual loci, edgeR-normalized DNase-seq tracks were produced converting BAM files into BedGraphs using BEDtools 2.24.0 genomecov with −*bg* and −scale parameters (Quinlan, 2014). Sizes of the libraries were obtained from the dba.analyse object (bFullLibrarySize = TRUE). Scaling factors were calculated by dividing the minimum library count by the size of each individual library. Bigwig files were compiled from normalized BedGraphs using bedGraphToBigWig v4. Coverage of individual loci were plotted in R3.5.0 using the wiggleplotr 1.6.1 R package.

### Assigning DHSs to closest TSS

DHSs were mapped to the closest TSS using the chipenrich v2.6.1 R package (Welch et al, 2014) and the predefined locus definition nearest_tss for the mm10 genome.

### STRING-db sub-network enrichment analysis

The STRINGdb v1.22.0 R package (Franceschini et al, 2013) was used to identify enriched protein–protein interaction sub-networks among predefined gene sets. The complete network (version 10) for the *Mus Musculus* (10,090) organism was selected and only gene symbol that mapped to STRINGdb IDs were retained. Sub-networks were identified using the Walktrap community-finding algorithm implemented in igraph v.1.2.4.

### Total RNA extraction and sequencing

25–50 mg of frozen liver material was used for total RNA extraction using 1 ml TRIzol reagent (cat. no. 15596026; Thermo Fisher Scientific). Samples were homogenized using a MagNA Lyser homogenizer (Roche) using two runs of 30 s at 6,500 rpm. Aliquots of ~600 $\mu$l of homogenized material were transferred into pre-spun 2-ml Phase Lock Gel-Heavy tubes (5Prime) and incubated for 5 min at room temperature. 100 $\mu$l chloroform–isoamyl alcohol was added to each aliquot. The tubes were shaken vigorously for 15 s and incubated for 2 min. The tubes were centrifuged at 12,000$g$ for 10 min at 4°C allowing phase separation. Sample-matching upper aqueous phases were pooled into new pre-spun 2-ml Phase Lock Gel-Heavy tubes and additional 200 $\mu$l chloroform–isoamyl alcohol was used to perform a second phase separation. Aqueous phases were then measured and transferred into fresh tubes where RNA precipitation with 0.5 ml isopropyl alcohol was performed. After 10-min incubation, the samples were centrifuged at 12,000$g$ for 10 min at 4°C. The precipitated RNA was washed once by using 1 ml 75% ethanol before resuspension in EB buffer (QIAGEN). The samples were NanoDrop-quantified and the RNA integrity number measured using an Agilent RNA 6000 Nano Kit (cat. no. 5067-1511; Agilent Technologies) on a BioAnalyzer instrument. Samples with RNA integrity number above 8 were used for sequencing. Illumina libraries were prepared with the TruSeq Stranded Total RNA Sample Preparation kit with Ribo-Zero Gold from 100 or 150 ng of input RNA using 13 or 12 rounds of PCR for amplification, respectively. We generated PE 76-bp reads on a HiSeq 2500 using v4 reagents (Illumina).

### RNA-seq data processing

Libraries were demultiplexed and then passed to the STAR v2.4.0f1 aligner (Dobin et al, 2013) to map reads to the UCSC reference genome mm10. The resulting bam files were sorted using SAMtools. Bam files of the same biological replicate from different sequencing runs were merged and filtered using SAMtools with options -q 255 and -F 512. To quantify gene expression, reads overlapping genes in the UCSC RefSeq gene annotation were counted using the summarizeOverlap function (with parameters mode = "Union," single-End = FALSE, ignore.strand = FALSE, and fragments = TRUE) of the GenomicAlignments (v1.6.3) R package.

### Differential gene expression analysis on RNA-seq data

EdgeR (v.3.12.1) (Robinson et al, 2010; McCarthy et al, 2012) was used to perform differential expression analysis on the count matrix. Only genes that show CPM ≥ 0.4 in at least three samples per treatment-strain group in at least one group were retrained for analysis. Normalization factors to scale for library size were calculated using the trimmed mean of M values (TMM) method. The design is based on assigning each sample to a treatment-strain group. The glmQLFit and glmQLFTest methods were used to fit a quasi-likelihood negative binomial generalized log-linear model and to perform gene-wise tests for differential expression. Differentially expressed genes were identified as genes showing at least |log2 FC| ≥ 0.58 between selected contrasts and a FDR ≤ 0.01.

### TFBS enrichment

We searched the different groups of DHSs for enrichment of TF binding sites using the HOMER software (Heinz et al, 2010). We performed both, de novo motif analysis and enrichment analysis of known TF motifs and overlapped the results to strengthen the robustness of the identified motifs.

For the enrichment analysis, we obtained known TF binding motifs from the JASPAR database (Khan et al, 2018). Because motifs of different TFs can be very similar, we used the root motifs from the TF motif clustering provided in the JASPAR database as input for HOMER. The score cutoffs to call binding sites were optimized for each root motif by running HOMER's findMotifs on all DHSs and taking the score that defined the 88% quantile based on the resulting score distribution. Finally, we ran HOMER's findMotif on the individual DHS groups providing the fasta files of all DHS regions for background parameter calculation and obtained enrichment and de novo analysis results. The Regulatory Sequence Analysis Tools (RSAT) (Nguyen et al, 2018) compare-matrices tool with default values was used to correlate de novo analysis-derived TF motifs with the known root motifs. We overlapped the results by assigning each known root motif the de novo motif with highest correlation.

### Protein extraction, concentration measurement, and immunoblotting

Liver tissue was dissociated in ice-cold Radioimmunoprecipitation assay (RIPA) buffer (89900; Pierce) complemented with proteases and phosphatases inhibitors (P8340, P2850, and P5726-1/100 each; Sigma-Aldrich) using the Covaris ultrasound homogenizer. Liver lysates were centrifuged (15 min, 16,000$g$, 4°C) and supernatants collected and stored at −80°C. SDS–PAGE samples were prepared in NuPAGE 4× buffer (NP0007; Invitrogen) and 10× reducing agent (NP0004; Invitrogen) or in 6× loading buffer (60 mM Tris, pH 6.8, 2% SDS, 9.8% glycerol, bromophenol blue, 0.6 mM DTT). Samples were heated (10 min, 70°C), centrifuged (3 min, 14,000$g$), and resolved on Invitrogen NuPAGE Bis-Tris gels (4–12% or 8% gel, MES running buffer, 160 V, 1 h). The gels were transferred to nitrocellulose membranes (20 V, 7 min) using the I-Blot system (#IB1001EU; Invitrogen). Membranes were blocked for 1 h in Odyssey Blocking Buffer (LI-COR 927-40000)/PBS (1:1) and subsequently incubated overnight at 4°C with primary antibodies (anti-Src, CST2123, rabbit polyclonal IgG, 1:1,000; anti–E-Cadh, CST3195, rabbit polyclonal IgG, 1:1,000; anti-actin, A5060; Sigma-Aldrich, rabbit polyclonal IgG, 1:15,000) diluted in Odyssey Blocking Buffer/PBS-Tween 0.1% (1:1). After washing three times with PBS-Tween 0.1%, the membranes were incubated for 1 h with secondary antibodies labelled with IRdyes diluted in Odyssey Blocking Buffer/PBS-Tween 0.1% (1:1). The membranes were washed (three times with PBS-Tween 0.1% and three times with PBS) and dried before scanning using the Odyssey Infrared Imaging System (LI-COR). Integrated intensities of the protein bands of each target were quantified in their respective fluorescent channels using Odyssey software (background subtraction top/bottom of quantified band). The integrated intensity for Src and E-Cadh were normalized to the integrated intensity for actin and displayed as % of control.

## List of software and databases.

| Software/Database/ R package | Reference/Web link |
| --- | --- |
| bcl2fastq | https://support.illumina.com/sequencing/sequencing_software/bcl2fastq-conversion-software.html |
| Picard | http://broadinstitute.github.io/picard/ |
| bamUtil | https://genome.sph.umich.edu/wiki/BamUtil |
| bamsignals | http://bioconductor.org/packages/release/bioc/html/bamsignals.html |
| bedGraphToBigWig | http://hgdownload.soe.ucsc.edu/admin/exe/linux.x86_64/ |
| wiggleplotr | http://bioconductor.org/packages/release/bioc/html/wiggleplotr.html |
| igraph | https://igraph.org |
| GenomicAlignments | http://bioconductor.org/packages/release/bioc/html/GenomicAlignments.html |
| STRING database | http://string-db.org/ |
| JASPAR motif database | http://jaspar.genereg.net/ |
| GeneCard database | https://www.genecards.org/ |
| Cutadapt | https://doi.org/10.14806/ej.17.1.200 |

## Data Deposition

All data generated in this manuscript are included in the NCBI GEO under the accession number GSE131344.

## Supplementary Information

## Acknowledgements

Technical sequencing help was given by S Dessus-Babus and E Oakeley. We also thank M Stadler, F Hahne, P Marc, P Couttet, B Peterson, S Hoersch, O Grenet, A Vicart, and S-D Chibout for scientific discussions. This work was supported by Innovative Medicine Initiative Joint Undertaking (115001) (MARCAR project: http://www.imi.marcar.eu/) and Novartis. C R Wolf was a recipient of a Cancer Research UK program grant C4639/A10822. J Perner, J Beil, J Zhu, A Del Río-Espínola, L Morawiec, M Westphal, V Dubost, M Altorfer, U Naumann, A Mueller, K Kapur, M Borowsky, J Moggs, and R Terranova are full time employees of Novartis. A Vitobello was a recipient of a Novartis Institutes for Biomedical Research Postdoctoral Fellowships.

### Author Contributions

A Vitobello: conceptualization, resources, data curation, software, formal analysis, supervision, funding acquisition, validation, investigation, visualization, methodology, project administration, and writing—original draft, review, and editing.

J Perner: conceptualization, resources, data curation, software, formal analysis, supervision, validation, investigation, visualization, methodology, project administration, and writing—original draft, review, and editing.

J Beil: conceptualization, resources, data curation, software, formal analysis, supervision, validation, investigation, visualization, methodology, project administration, and writing—original draft, review, and editing.

J Zhu: conceptualization, resources, data curation, software, formal analysis, supervision, validation, investigation, visualization, methodology, project administration, and writing—original draft, review, and editing.

A Del Río-Espínola: data curation, software, formal analysis, validation, visualization, and methodology.

L Morawiec: conceptualization, resources, data curation, software, formal analysis, supervision, funding acquisition, validation, investigation, visualization, methodology, project administration, and writing—original draft, review, and editing.

M Westphal: data curation, software, formal analysis, supervision, validation, investigation, visualization, and methodology.

V Dubost: data curation, software, formal analysis, supervision, validation, investigation, visualization, and methodology.

M Altorfer: data curation, software, formal analysis, supervision, validation, investigation, visualization, and methodology.

U Naumann: data curation, software, formal analysis, supervision, validation, investigation, visualization, and methodology.

A Mueller: resources, data curation, software, formal analysis, supervision, validation, investigation, visualization, and methodology.

K Kapur: conceptualization, resources, data curation, software, formal analysis, supervision, validation, investigation, visualization, methodology, and project administration.

M Borowsky: conceptualization, resources, data curation, software, formal analysis, supervision, investigation, visualization, methodology, and project administration.

C Henderson: conceptualization, resources, data curation, software, formal analysis, supervision, investigation, visualization, methodology, and project administration.

CR Wolf: conceptualization, resources, data curation, software, formal analysis, supervision, funding acquisition, investigation, visualization, methodology, project administration, and writing—original draft, review, and editing.

M Schwarz: conceptualization, resources, data curation, software, formal analysis, supervision, funding acquisition, validation, investigation, visualization, methodology, project administration, and writing—original draft, review, and editing.

J Moggs: conceptualization, resources, data curation, software, formal analysis, supervision, funding acquisition, validation, investigation, visualization, methodology, project administration, and writing—original draft, review, and editing.

R Terranova: conceptualization, resources, data curation, software, formal analysis, supervision, funding acquisition, validation, investigation, visualization, methodology, project administration, and writing—original draft, review, and editing.

## Conflict of Interest Statement

The authors declare that they have no conflict of interest.

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
