## [Reviewer comments · Life Science Alliance]

Life Science Alliance

Drug-induced chromatin accessibility changes associate with sensitivity to liver tumor promotion

Antonio Vitobello, Juliane Perner, Johanna Beil, Jiang Zhu, Alberto DelRio, Laurent Morawiec, Magdalena Westphal, Valerie Dubost, Marc Altorfer, Ulrike Naumann, Arne Mueller, Karen Kapur, Mark Borowsky, Colin Henderson, Roland Wolf, Michael SCHWARZ, Jonathan Moggs, and Remi Terranova

DOI: <https://doi.org/10.26508/lsa.201900461>

Corresponding author(s): *Remi Terranova, Novartis Institutes for Biomedical Research (NIBR)*

Review Timeline:

Submission Date:	2019-06-20
Editorial Decision:	2019-07-24
Revision Received:	2019-09-06
Editorial Decision:	2019-09-20
Revision Received:	2019-09-26
Accepted:	2019-09-26

Scientific Editor: Andrea Leibfried

Transaction Report:

July 24, 2019

Re: Life Science Alliance manuscript #LSA-2019-00461-T

Remi Terranova
Novartis Institutes for Biomedical Research (NIBR)

Dear Dr. Terranova,

Thank you for submitting your manuscript entitled "Xenobiotic-induced perturbations of open chromatin regulatory elements associated with sensitivity to liver tumor promotion". The manuscript has been evaluated by expert reviewers, whose reports are appended below.

As you will see, the reviewers find the analyses of genetic background-dependent sensitivity to phenobarbital interesting, and reviewer #3 supports publication of your work here. However, reviewer #2 and #3 raise strong concerns with the way the analyses were performed and do not think that your conclusions are sufficiently supported by the data provided. They note that the initially chosen thresholds are not stringent enough, affecting the whole analyses. Rev#2 furthermore thinks that the data could have been analyzed further to provide valuable insight into SNPs correlating with the genotype-dependent differences observed. We discussed your work in light of these comments and concluded that we cannot offer publication, as the outcome of addressing these issues and the value provided to others is too unclear at this stage. If you wish to expedite publication of the current data, it may be thus best to pursue publication at another journal.

Given the interest in the topic, we would be open to resubmission to Life Science Alliance of a significantly revised and extended manuscript that fully addresses the reviewers' concerns and is subject to further peer-review. Please note that priority and novelty would be reassessed at resubmission and that we would need strong support on such a revised version from reviewer #1 and #2 for publication here.

Regardless of how you choose to proceed, we hope that the comments below will prove constructive as your work progresses.

Thank you for thinking of Life Science Alliance as an appropriate place to publish your work.

Sincerely,

Reviewer #1 (Comments to the Authors (Required)):

In this paper, authors tend to reveal the underlying mechanisms that drive the different susceptibility of liver cancers caused by xenobiotic in two different mouse strains. They used C57BL/6 and B6C3F1 mice that have different genetic background and sensitivity to phenobarbital to reveal the difference of phenobarbital-treated livers at epigenetic and transcriptomic levels. Although the topic addressed by this paper is very interesting, the data analysis is poorly done and their conclusion is not convincing enough.

Following are the critical points that authors need to improve or discuss in their study. Points 1-3 are essential for the whole concept of this paper that should be addressed properly.

1. In Figure 2a, authors plotted shared, B6C3F1-unique or C57BL/6-unique -DHSs. However, from the pattern, one could easily see that B6C3F1-unique -DHSs were also changed in C57BL/6 with relatively smaller FC and less consistency between replicates. The same is true for C57BL/6-unique -DHSs. Authors should take more stringent threshold for this analysis.
2. In Figure 2b, authors showed that shared -DHSs tend to occur in regions of strongest DHS log₂ FC. This could be due to the high FC and high consistency between replicates which lead to statistically high significance. In general, the -DHSs are mostly consistent between two mouse strains. Authors should take more stringent threshold to define shared and unique -DHSs.
3. Point 2 & 3 should also be considered for RNA-seq data analysis. And -RNAs, compared with -DHSs, are even more consistent between two mouse strains.
4. Figure 3c showed the overlap between Δ -RNA and Δ -DHS. Why the overlap is so small? Does this suggest that Δ -DHS is not the cause of Δ -RNA?
5. The -DHSs on Src in Figure 4d is not that significant. Also, although Cdh1 and Src is associated with Wnt signaling, but their influence on Wnt signaling are different. Whether the -DHSs on Cdh1 and Src mediate the different Wnt activation in different mouse strain is questionable.
6. The analysis on Figure 4 & 5 are based on Figure 2 & 3 and should be improved according to the change of Figure 2 & 3.

Reviewer #2 (Comments to the Authors (Required)):

In this paper, the authors conducted DNase-seq and RNA-seq experiments using liver tissue with/without phenobarbital (PB) treatment from two different mouse strains (B6C3F1 and C57BL/6), respectively. They found mild strain-specific differences at both chromatin accessibility and transcription levels that may be associated with PB-mediated tumor promotion sensitivity. These strain-specific differences included a few genes in the beta-catenin pathway which has been reported to contribute to PB-mediated liver tumor promotion. The authors also predicted the potential transcription factors that may play a role upon PB treatment.

Overall, I think this paper is interesting and will help the community better understand the molecular basis of PB-mediated tumor promotion sensitivity. In my opinion this paper is suitable for LSA journal but there are a few issues that need to be addressed.

1. Although the DHS signals show a high degree of reproducibility across samples, the samples from the same group are not clustered together (Supplementary Figure S2b), which indicates the effect of PB treatment is not distinguishable from that of vehicle treatment. This is especially true for C57BL/6 samples.

- 2.It would be better to also perform similar analysis on RNA-seq data as Figure S2b showed.
- 3.Heatmap in figure 3a showed that part of the strain unique genes seem to be better grouped as the "shared" genes. Different cutoffs need to be used to determine the relative portion of the shared and unique strain effects.
- 4.For figure 3c, whether other methods assigning DHS to genes will increase the overlap between DHS and RNA changes ?
- 5.For figure 4a/b, it will be interesting to check if there are SNPs inside the peaks with strain specific effect, and which TF binding was possibly involved. It will be much better to systematically check the SNP density in strain specific or shared DHS peaks and see the differences.
- 6.For figure 5b, were the gene expression level of TFs under consideration?
- 7.Are C57BL/6J, C57BL/6 and C57BL6/N the same mouse strain? If so, it should be used consistently throughout the text.
- 8.Please cite properly. e.g. EdgeR method was not cited in this manuscript.

Reviewer #3 (Comments to the Authors (Required)):

In this present manuscript, the authors have addressed using well-defined functional genomic strategies a very interesting and important question regarding the susceptibility differences in the pathogenesis of hepatocellular carcinoma using xenobiotic-induced mouse model. The experimental approaches and analysis are strong and support the conclusions drawn. This investigation is bound to interest a broad scientific community. This manuscript should be considered for publication with a few textual modifications.

Minor comment: The results and the discussion section can be made more lucid to explain the broad significance to scientists and clinicians who are not proficient with changes in DNA structure, epigenetic modifications and the concept of 'distal enhancers' and their contributions to 3-dimensional genome architecture and its effect on transcription..

Reviewer #1 (Comments to the Authors (Required)):

In this paper, authors tend to reveal the underlying mechanisms that drive the different susceptibility of liver cancers caused by xenobiotic in two different mouse strains. They used C57BL/6 and B6C3F1 mice that have different genetic backgrounds and sensitivity to phenobarbital to reveal the difference of phenobarbital-treated livers at epigenetic and transcriptomic levels. Although the topic addressed by this paper is very interesting, the data analysis is poorly done and their conclusion is not convincing enough. Following are the critical points that authors need to improve or discuss in their study. Points 1-3 are essential for the whole concept of this paper that should be addressed properly.

We thank the reviewer #1 for constructive comments and appreciate his/her acknowledgement of the value of the addressed topic. The impact of regulatory genome variations in disease/phenotype susceptibility has been increasingly acknowledged and characterized over the last few years including through the systemic integration of GWAS and chromatin structure data. Mutations occurring in cancer often misregulate enhancers that normally control the signal-dependent expression of growth-related genes. This misregulation can result from *trans*-acting mechanisms, such as activation of the transcription factors or epigenetic regulators that control enhancer activity, or can be caused in *cis* by direct mutations that alter the activity of the enhancer or its target gene specificity (1,2). Such impact of regulatory genome genetic and epigenetic variations is likely to account for inter-individual phenotypic variations. Yet few reports have systematically explored that phenomenon in *in vivo* settings.

In this study we have exploited established mouse strain-specific differences in sensitivity to phenobarbital-mediated liver tumor promotion to explore early molecular events and strain differences that may account for differential phenotypic outcome. We specifically focus on profiling chromatin accessibility as a readout to the regulatory genome anchored to transcriptional expression changes.

We were disappointed to hear that the reviewer finds the analysis is *poorly done* and not supportive of the conclusions. From the available comments, we understand this is based on the perception that inappropriately relaxed statistical thresholds were used in the reported analyses supporting strain-selective DHS and RNA changes upon PB treatment. To our understanding, this perception appears to be based on the visual representations of differential strain effects in the original figures 2 and 3, where changes reported as '*unique*' to a strain were also visually apparent (with lower FC and consistence as noted by the reviewer) in the other strain in the original head-to-head heatmap representations.

We fully agree with the reviewer's perception that indeed, this study reports quantitative rather than qualitative differences in DHS and expression; we did design the figures and acknowledged that trend in the original submission. We propose that this consistent trend is expected from the investigated biological system, and to avoid any misperception, would like to offer a point-by-point response to the criticisms.

Further acknowledging that this important point needs to be consolidated to help the reader interpret the results, we have now made significant data illustration and supporting text changes to support our position below. We specifically agree that the use of '*unique*' or '*specific*' does not properly describe the nature of the strain effects, which rather account for differential *magnitude* or *enrichment* of effects. We have adapted the semantic throughout.

We sincerely hope that these revisions will address the reviewers' concerns regarding the statistical thresholds applied within the manuscript.

1. In Figure 2a, authors plotted shared, B6C3F1-unique or C57BL/6-unique -DHSs. However, from the pattern, one could easily see that B6C3F1-unique -DHSs were also changed in C57BL/6 with relatively smaller FC and less consistence between replicates. The same is true for C57BL/6-unique -DHSs. Authors should take more stringent threshold for this analysis.
2. In Figure 2b, authors showed that shared -DHSs tend to occur in regions of strongest DHS log₂ FC. This could be due to the high FC and high consistence between replicates which lead to statistic high significance. In general, the -DHSs are mostly consistent between two mouse strains. Authors should take more stringent threshold to define shared and unique -DHSs.
3. Point 2 & 3 should also be considered for RNA-seq data analysis. And -RNAs, compared with -DHSs, are even more consistent between two mouse strains.

We propose to address points 1-3 together as they all relate to statistical threshold selection and choices of differential DHS/RNA data visualization that overall point to qualitatively consistent, quantitatively differential, effects across strains as noted by the reviewer.

- In this study we have used consistent statistical thresholds ($\text{Log}_2\text{FC} \geq 0.58$ and $\text{FDR} \leq 0.01$) for all analyses on both strains. Using those thresholds, we did report common signatures and strain-enriched signatures, which we acknowledged as being of a

quantitative nature through providing head-to-head heatmap representations of the data as noted by the reviewer. This was originally stated in the original results section for DHS (Figure 2) “Comparing DHS heatmaps head-to-head points to consistent trends in chromatin accessibility changes in both strains (albeit only reaching the statistical thresholds in one), plausibly consistent with differences in tissue composition/PB effects”, as well as for RNA changes (Figure 3) “comparing B6C3F1 vs C57BL/6 effects showed that the shared effects tended to occur in regions of strongest gene expression changes (Figure 3b), and overall a trend of quantitative rather than qualitative gene expression differences was apparent”

- We now further reinforce this conclusion through adapting the Figures 2 and 3, providing volcano plot illustrations indicating statistical thresholds supporting the characterization of strain-enriched effects, while maintaining the head-to-head heatmap representations to acknowledge the consistency of effects across strains.
 - We have also adapted the description of the effect, avoiding the use of ‘unique’ and rather using ‘enriched’ (all figures adapted accordingly), ‘quantitative’ or ‘selective’.
 - We also further elaborate the limitations of the study in the discussion section. We cannot fully exclude that qualitative effects take place but they would require approaches beyond the scope of this study (e.g single-cell approaches).
- The consistency of global molecular changes is expected in this experimental set-up and working on total liver tissue.
- Liver phenotype: As highlighted in the historical tumor incidence **Table S1** and references within, the tested strains C57BL/6 and hybrid B6C3F1 are relatively resistant or responsive to PB-mediated liver tumor promotion. B6C3F1 show variable incidence of tumor formation across studies and C57BL/6 can also develop beta-catenin mutated tumors upon DEN initiation on long-term PB treatment. Thus, while anchoring the set-up on differential liver tumor formation susceptibility, we still work and compare liver tissue from mice, at an early stage of chronic PB treatment (3 months), and in absence of reported phenotypic differences as noted in the manuscript. From this set-up and large amount of historical data, strong qualitative differences are not expected from total liver molecular profiling.
- Very stringent statistical cut-offs would artificially hide biologically relevant effects in the context of complex tissue responses to drug treatment, and likely result in enhancing the detection of xenobiotic metabolism effects at the expense of pathways underlying the early tumorigenic response.
- Liver tissue composition: While considered a very homogeneous tissue, the actual cellular composition of the liver remains poorly understood. Recent single-cell RNA sequencing of about 10,000 cells have highlighted the complexities of cellular sub-types (3,4).
 - Liver tissue PB response: In addition, we previously reported that a subset of perivenous, Wnt/beta-catenin dependent, Dlk1-Dio3 locus non-coding RNA expressing cells, may contribute liver tumor promoting cellular/molecular signature at early stages of PB treatment (5). These cells only represent a small fraction of the liver tissue (see Lempiainen et al, Fig 4 illustration below – Dlk1-Dio3 ncRNA signals are indicated with arrows) and this signature was identified using similar statistical thresholds (> 1.5-fold change, adjusted p-value < 0.01).

FIG. 4. Expression and localization of *Meg3* in the liver. Upper panel: Colocalization of *Meg3* transcript (ISH: blue) and GS (IHC: brown) protein in control (91 days) and treated (1, 14, and 91 days) livers. Boxes indicate venous regions of the central zone of the lobule that are magnified in the middle and bottom panels. Middle panel: Localization of *Meg3*. Arrows indicate cells showing positive *Meg3* ISH staining. Blue staining at day 1 is unspecific non-nuclear background. Bottom panel: Colocalization of *Meg3* and GS stainings. Central vein is indicated by "CV."

We are thus overall looking at subtle and early changes driven by a drug in animals and working on total liver samples from complex liver/PB-effect tissue. We have based the RNAseq and DNase-seq analyses thresholds on historical and practical experience and accounting for the complexities of the experimental tissue/system. We would be wary of misleading the future readership that strain-specific phenotypic outcome differences should be supported by qualitative rather than quantitative molecular effects in these experimental conditions and would not expect delivering enhanced biological interpretation by selecting more stringent thresholds. Finally all raw data generated in this manuscript are included in NCBI GEO under the accession number GSE131344 and available for further analysis and integration by future readers/researchers.

We hope this overall addresses the primary concerns of the reviewers and editors. We have made substantial changes in the figures displays and supporting manuscript sections and believe this enabled significant improvement of the manuscript. We welcome further comments to enhance the manuscript further.

4. Figure 3c showed the overlap between Δ -RNA and Δ -DHS. Why the overlap is so small? Does this suggest that Δ -DHS is not the cause of Δ -RNA?

We thank the reviewer for this comment. To provide a comprehensive answer we would first like to provide an additional background on the scope of the DHS and RNA functional integration. Many coding genes are well annotated with their biological functions. Non-coding regions, such as those we identify through open chromatin profiling however typically lack such annotation. There are different (computational and experimental) approaches that can be used to assign biological meaning to a set of non-coding genomic regions such as DHSs. In this manuscript we explore two separate computational approaches:

1. We **assigned the nearest TSS** (regardless of genomic distance) to each delta-DHS (revised figure 2d): this analysis as all such genomic proximity-based functionalization of

non-coding landscape is imperfect but enables exploring possible functional effects of the reported non-coding DHS landscape, with the assumption that changes in the proximo-distal chromatin landscape may impact expression levels of nearby genes.

2. Since the vast majority of identified delta-DHS do not account for proximal change in gene expression, as elaborated in the manuscript, we have also investigated the possibility that this delta-DHS landscape may be **enriched in transcription factor motifs (Figure 5)** and represent functionally relevant regulatory landscape changes that may not immediately account for gene expression changes.

To the first approach and specific reviewer #1's question on Delta-DHS and Delta-RNA overlap:

- Our data show that the majority of delta-DHS effects do not occur at proximity to the TSS and are on average located ≥ 45 kbp away to a TSS (revised Figure 2e, Figure 1c, Supplementary Figure S2b,c). The delta-DHS landscape thus rather encompasses distal regulatory regions, presumably including enhancer elements. While an important regulatory component of spatio-temporal gene expression regulation, we do not a priori expect a strong correlation between this chromatin landscape and gene expression changes. No single epigenomic or chromatin-based profiling readout is actually reported as predictive to gene expression changes; rather, a combination of readouts is usually necessary to provide insights into gene regulatory events associated with changes in mRNA expression. Thus Δ -DHS is not necessarily directly the cause of Δ -RNA and rather a complementary landscape that enables identification of discrete changes not necessarily associated with immediate transcriptional changes. We have made consistent results section changes to highlight this limited overlap in the revised manuscript.

However, and as also pointed by reviewer #2, we did use a single method to assign DHS to genes in the original manuscript. Acknowledging both comments we have now deployed alternative methods to explore the variation on DHS/RNA changes overlap. The new analyses are reported in a new **Supplementary Figure S5** and in the results section. We provide a summary of this comparison below.

- In the original analysis (revised Figure 3d), using the **nearest TSS** calling method, we find that 29 out of 167 (17%) differentially expressed genes in B6C3F1 and 3 out of 25 (12%) differentially expressed genes in C57BL/6N show this potential functional overlap. This method considers only each delta-DHS is associated to exactly one delta-RNA.
- In the two newly reported association approaches, multiple assignments of a delta-DHS to delta-RNA and vice versa are allowed. In the first newly reported **TSS-centric** approach we account for the situation that a delta-RNA can be regulated by multiple DHS in its proximity regardless of whether it is the closest TSS to the DHS. Thus, we now explore whole-genome TSS ($n=34,219$) for flanking delta-DHS association in windows of $\pm 5, 10, 50$ or 100 kbp. Looking specifically at the overlap between the delta-RNA landscape and flanking delta-DHS in each strain we report increasing overlap, from 16 out of 167 (10%, in ± 5 kbp settings) and up to 44 out of 167 (26% in ± 100 kbp settings) in B6C3F1 and 4% (± 5 kbp) to 24% (± 100 kbp) overlap in C57BL/6N. Thus, using a range of genomic distances modulates, but does not overall significantly change the proportion of the overlap compared to the originally reported nearest TSS method.
- In a second complementary **DHS-centric** approach, we account for the situation that each delta-DHS could regulated any TSS in its proximity. Similar to the TSS-centric approach, we have explored the $\pm 5, 10, 50$ and 100 kbp flanking regions of each delta-DHS in both strains. Since there are more delta-DHS than delta-RNA effects and the majority of DHS are located distally to TSS, the overlap output appears lower than

scanning from the TSS, ranging from 4% to 13% in +/- 5 to +/-100 kbp in B6C3F1 and close to null in C57BL/6N.

Thus overall we propose that a small but consistent proportions (10-26% depending on conditions) of delta-RNA have a delta-DHS within a functional genomic range. On the other hand a low proportion of delta-DHS (more numerous and overall distally located) have a linkable delta-RNA locus at proximity.

These analyses are standard in the field of non-coding landscape functionalization but are acknowledgeably limited in scope. The deployment of Chromosome Conformation Capture (3C)-based methods would be necessary to more robustly associate chromatin/gene expression changes and interrogate these functional interactions in an unsupervised manner, but this is currently outside the scope of this study. These limitations are highlighted in the discussion section.

The functionalization of the DHS landscape beyond direct effects on transcription is actually core to this manuscript and consistent with historical investigations of early molecular drivers PB-induced of liver tumor promotion (5-7). Epigenetic changes may take place at early stage of carcinogenesis (8) in absence of associated transcriptional effects. Here we have investigated the overall delta-DHS landscape through different lenses, analyzing it in terms of genetic information and enrichment for transcription factor motifs as described above and in figure 5 of the manuscript.

5. The -DHSs on *Src* in Figure 4d is not that significant. Also, although *Cdh1* and *Src* is associated with Wnt signaling, but their influence on Wnt signaling are different. Whether the -DHSs on *Cdh1* and *Scr* mediate the different Wnt activation in different mouse strain is questionable.

The delta-DHS in the *Src* locus visualized in the Figure 4 are significant from a statistical viewpoint and using the reported cut-offs ($\text{Log}_2 \text{FC} \geq 0.58$, $\text{FDR} \leq 0.01$). The actual data is available from Supplementary Table S3. We have modified the legend to highlight that light grey boxes represent peaks beyond selected cut-offs.

We have highlighted those loci as they were both showing greater magnitude of changes in B6C3F1 at both DHS and RNA levels, and are both directly or indirectly connected to Wnt/ β -catenin signalling, previously reported as one of the key growth control pathway affected upon PB treatment. In this section, we specifically explored whether RNA and DHS changes translate at the functional, protein expression level. We show so using western-blot analysis. We however do not comment or speculate on whether *Cdh1* or *Src* may mediate differential degrees of Wnt activation in different mouse strains.

We agree that using different and significantly more stringent thresholds will potentially hide these signatures, for the least *Src*. The choice of threshold, acknowledgement of the complexities of the tissue, cell-of-origin of the molecular changes in response to the drug and to the emerging early carcinogenicity signature is a set of key challenges working from *in vivo* samples. These are discussed above in response to comments 1-3 and further elaborated in the manuscript. We welcome additional perspectives from the reviewer.

6. The analysis on Figure 4 & 5 are based on Figure 2 & 3 and should be improve according to

the change of Figure 2 & 3.

We agree that using different thresholds will change the overall set of signatures. This comment is related (and dependent) to above comments. We have changed the figures 2 and 3 to further highlight the cross-strain trends of effects and the quantitative nature of changes inherent to the experimental system and background biology (above comments).

Reviewer #2 (Comments to the Authors (Required)):

In this paper, the authors conducted DNase-seq and RNA-seq experiments using liver tissue with/without phenobarbital(PB) treatment from two different mouse strains (B6C3F1 and C57BL/6), respectively. They found mild strain-specific differences at both chromatin accessibility and transcription levels that may be associated with PB-mediated tumor promotion sensitivity. These strain-specific differences included a few genes in the beta-catenin pathway which has been reported to contribute to PB-mediated liver tumor promotion. The authors also predicted the potential transcription factors that may play role upon PB treatment. Overall, I think this paper is interesting and will help the community better understand the molecular basis of PB-mediated tumor promotion sensitivity. In my opinion this paper is suitable for LSA journal but there are a few issues that need to be addressed.

We are pleased the reviewer finds the manuscript and data interesting, relevant to the community and overall suitable for publication in the LSA journal. We would like to address the reviewer's comments through a point-by-point response below.

1. Although the DHS signals show high degree of reproducibility across samples, the samples from the same group are not clustered together (Supplementary Figure S2b), which indicates the effect of PB treatment is not distinguishable from that of vehicle treatment. This is especially true for C57BL/6 samples.

Indeed the clustering analysis indicates that the PB effect is not the strongest variance-introducing factor. Performing a PCA analysis on the top 5000 most variable DHS sites (added to Supplementary Figure S3b) shows that a DHS signature related to PB treatment is only the third strongest source of variation in this data. The strongest source of variation is the strain background, as rightfully picked up by the cluster analysis. The source contributing to the second PC and to the mixing of vehicle and PB treated samples in the clustering remains unknown. We hypothesized that experimental variations (DNase I digestion, library preparation, size selection and sequencing) or stochastic and biological fluctuations of individual DHS sites could have affected this data. However, with the supervised differential analysis presented in the manuscript, we home in on the PB-treatment mediated effects that were picked-up in the unsupervised PCA analysis. This is also illustrated by higher contributions of the shared and strain-enriched delta-DHS peaks towards PC3 (see Figure below showing the PC3 loadings for shared and strain-enriched delta-DHS, as well as for all DHS used in the PCA analysis).

The clustering, PCA and potential experimental/tissue variability accounting for lack of full treatment clustering is now reported in the second results section

2. It would be better to also perform similar analysis on RNA-seq data as Figure S2b showed.

We agree that for homogeneity of data representation it would be valuable to provide similar clustering analysis for the RNA-seq data. We have used the top 1000 most variable genes to correlate RNAseq data across samples and provide this data in a new Supplementary Figure S3c. In contrast to the DHS, the samples from each strain and treatment groups are clustered together for both B6C3F1 and C57BL/6N. This is also supported by PCA that separates the samples by strain and treatment based on PC1 and PC2, which together account for 72% of the variance in the data.

We noted that the PB-treatment effect is more pronounced in the unsupervised analysis of the RNA than of the DHS peaks, pointing to smaller PB-induced effect sizes in DHS compared to RNA changes. Consistently, we find that most DHS peaks are pre-existing at baseline with PB treatment leading to subtle increases or decreases in accessibility. On the other hand, the range of gene expression changes are much broader, overall supporting observed differences in clustering and PCA.

3. Heatmap in figure 3a showed that part of the strain unique genes seem to be better grouped as the "shared" genes. Different cutoffs need to be used to determine the relative portion of the shared and unique strain effects.

We thank the reviewer for this observation, part of which is reminiscent to the reviewer#1 points on data analysis thresholds and visualization.

If we understand the specific question here, while the reviewer 2 acknowledges that the strain-specific effects may be of a quantitative nature (and thus represent a relative 'mild strain-specific difference'), he questions whether different cut-offs may help enhance the difference between strain-specific and shared effects.

As discussed above we fully expect that quantitative rather than qualitative effects may support the reported difference in tumor development susceptibility. While the heatmaps support this consistency trend, we adapted the visualization of data based on this and reviewer 1 input. We have now included volcano plots in both Figure 2 and 3 to better illustrate the output of the statistical test. As noted above we have also adapted the reporting of the effects to avoid the use of 'strain-unique' effects. We hope this clarifies and will enable transparent interpretation of the data for the readership.

4. For figure 3c, whether other methods assigning DHS to genes will increase the overlap between DHS and RNA changes ?

This point is also equivalent to the reviewer#1 point 4. We refer to this response, highlighting some background on DHS functionalization and proposing two separate methods, TSS-centric and DHS-centric to investigate whether other DHS-gene assignment approaches may change the overlap between DHS and RNA. This data is reported in a new Supplementary Figure S5 and discussed in the results section.

5. For figure 4a/b, it will be interesting to check if there are SNPs inside the peaks with strain specific effect, and which TF binding was possibly involved. It will be much better to systematically check the SNP density in strain specific or shared DHS peaks and see the differences.

We agree with the reviewer that connecting the SNPs to strain specific effects would be an interesting analysis to perform. As pointed out by the reviewer, answering the question of how individual strain-specific SNPs might disrupt TF binding and the DHS peak strength is difficult with the data at hand, mostly because other strain-specific or technical effects could confound the genetic variation in a single SNP analysis. A dedicated (quantitative-trait loci) study comprising sufficient individuals to disentangle the genetic component of individual SNPs from potentially confounding factors will be much better suited to answer this question.

However, we were able to test the hypothesis that unique/enriched delta DHS peaks might be enriched in SNPs. To do so, we overlapped the DHS peaks with the known strain-distinguishing SNPs from the Mouse Genomes Project (also used for genome imputation, see methods). Of the 69871 SNPs in DHS peaks, 304 SNPs were in delta-DHS peaks. We did not observe enrichment of SNPs in strain-enriched delta DHS (~23%, 235 out of the 1037 DHS sites) compared to the shared delta-DHS sites (~22%, 69 out of 319 DHS). When normalizing to the length of the DHS peaks, we observe a SNP frequency of 2.02 SNPs/kbp in strain-enriched delta-DHS and 1.79 SNPs/kbp shared delta-DHS. This is slightly above for the strain-enriched and slightly below for shared regions compared to the background frequency of 1.85 SNPs/kbp using all consensus DHS or a distance-to-TSS distribution aligned random background set. Also based on the number of SNPs per DHS site we could not identify a strong difference between enriched (on average 0.66 SNPs per peak) and shared (on average 0.60 SNPs per peak) delta DHS sites. This suggests a more complex genotype to phenotype relationship, affecting the regulatory program at different levels.

6. For figure 5b, were the gene expression level of TFs under consideration?

All JASPAR motif clusters were used for the enrichment analysis. The transcription factors listed in column “Genes” in Figure 5b are restricted to genes that have at least 100 RNA-seq read counts summed across all samples. Finally, the genes for each cluster are ordered by average expression across samples. We have clarified this selection criteria in the figure legends of Figure 5b and Supplementary Table S6.

7. Are C57BL/6J, C57BL/6 and C57BL6/N the same mouse strain? If so, it should be used consistently throughout the text.

Thank you for highlighting the need for consistency of mouse strain nomenclature. Many sub-strains of C57BL/6 exist and ‘C57BL/6’ represent the generic label for the strain. We highlighted in the legend of Table S1 that sub-strain differences exist and may represent one of the variable across historical *in vivo* liver tumor promotion studies: C57BL/6N (MGI:2159965) and C57BL/6J (MGI:3028467) are thus slightly different, so are C3H/HeN (MGI:2160972) and C3H/He (MGI:2159866).

In our *in vivo* studies, C57BL/6 mice were obtained from Taconic (Germany) and are specifically of C57BL/6N background. This sub-strain is identical to the females used for the generation of B6C3F1 obtained from Charles River.

We have now clarified the sub-strain in the methods and adapted the sub-strain name throughout the revised text and figures.

Many Substrains of C57BL/6 Exist

8. Please cite properly. e.g. EdgeR method was not cited in this manuscript.

We have now revised and/or included new citations, as well as a summary table of all softwares and databases used in this study.

Reviewer #3 (Comments to the Authors (Required)):

In this present manuscript, the authors have addressed using well-defined functional genomic strategies a very interesting and important question regarding the susceptibility differences in the pathogenesis of hepatocellular carcinoma using xenobiotic-induced mouse model. The experimental approaches and analysis are strong and support the conclusions drawn. This investigation is bound to interest a broad scientific community. This manuscript should be considered for publication with a few textual modifications.

Minor comment: The results and the discussion section can be made more lucid to explain the broad significance to scientists and clinicians who are not proficient with changes in DNA structure, epigenetic modifications and the concept of 'distal enhancers' and their contributions to 3-dimensional genome architecture and its effect on transcription..

We thank the reviewer for acknowledging his strong interest in the scientific question and in the quality of the experimental approaches and analyses.

We appreciate the comment on simplifying and clarifying some of the results and discussion concepts for the benefits of non-specialist readership. We have revised the manuscript in details to ensure smooth readability, including of expert epigenetic concepts. All changes are available from the track change version in this new submission.

We hope these edits and responses to the points raised by the reviewers #1 and #2 further enhances the quality and readability of the manuscript and will overall represent a study of interest to the broader community.

References

1. Maurano, M.T., Humbert, R., Rynes, E., Thurman, R.E., Haugen, E., Wang, H., Reynolds, A.P., Sandstrom, R., Qu, H., Brody, J. *et al.* (2012) Systematic localization of common disease-associated variation in regulatory DNA. *Science*, **337**, 1190-1195.
2. Sur, I. and Taipale, J. (2016) The role of enhancers in cancer. *Nat Rev Cancer*, **16**, 483-493.
3. Segal, J.M., Kent, D., Wesche, D.J., Ng, S.S., Serra, M., Oules, B., Kar, G., Emerton, G., Blackford, S.J.I., Darmanis, S. *et al.* (2019) Single cell analysis of human foetal liver captures the transcriptional profile of hepatobiliary hybrid progenitors. *Nature communications*, **10**, 3350.
4. Aizarani, N., Saviano, A., Sagar, Mailly, L., Durand, S., Herman, J.S., Pessaux, P., Baumert, T.F. and Grun, D. (2019) A human liver cell atlas reveals heterogeneity and epithelial progenitors. *Nature*, **572**, 199-204.
5. Lempiainen, H., Couttet, P., Bolognani, F., Muller, A., Dubost, V., Luisier, R., Del Rio Espinola, A., Vitry, V., Unterberger, E.B., Thomson, J.P. *et al.* (2013) Identification of Dlk1-Dio3 imprinted gene cluster noncoding RNAs as novel candidate biomarkers for liver tumor promotion. *Toxicol Sci*, **131**, 375-386.
6. Thomson, J.P., Lempiainen, H., Hackett, J.A., Nestor, C.E., Muller, A., Bolognani, F., Oakeley, E.J., Schubeler, D., Terranova, R., Reinhardt, D. *et al.* (2012) Non-genotoxic carcinogen exposure induces defined changes in the 5-hydroxymethylome. *Genome biology*, **13**, R93.
7. Thomson, J.P., Hunter, J.M., Lempiainen, H., Muller, A., Terranova, R., Moggs, J.G. and Meehan, R.R. (2013) Dynamic changes in 5-hydroxymethylation signatures underpin early and late events in drug exposed liver. *Nucleic acids research*, **41**, 5639-5654.
8. Feinberg, A.P., Ohlsson, R. and Henikoff, S. (2006) The epigenetic progenitor origin of human cancer. *Nat Rev Genet*, **7**, 21-33.

September 20, 2019

RE: Life Science Alliance Manuscript #LSA-2019-00461-TR

Dr. Remi Terranova
Novartis Institutes for Biomedical Research (NIBR)
Klybeckstrasse 141
Basel 4057
Switzerland

Dear Dr. Terranova,

Thank you for submitting your revised manuscript entitled "Drug-induced chromatin accessibility changes associate with sensitivity to liver tumor promotion". As you will see, reviewer #2 appreciates the introduced changes and we would thus be happy to publish your paper in Life Science Alliance pending final revisions necessary to meet our formatting guidelines.

- please provide table S1 as either word or excel file
- please add callouts in the manuscript text to figure 1E, 3D, 4A, B, D, E

A. FINAL FILES:

B. MANUSCRIPT ORGANIZATION AND FORMATTING:

Sincerely,

Reviewer #2 (Comments to the Authors (Required)):

In this manuscript, the authors conducted DNase-seq and RNA-seq experiments using liver tissue

with/without phenobarbital treatment from two different mouse strains, respectively. They found quantitative differences between two mouse strains at both chromatin accessibility and gene expression levels that may be associated with PB-mediated tumor promotion sensitivity. The revised manuscript address my concerns appropriately, I am satisfied with the improvement made by the authors.

September 26, 2019

RE: Life Science Alliance Manuscript #LSA-2019-00461-TRR

Dr. Remi Terranova
Novartis Institutes for Biomedical Research (NIBR)
Klybeckstrasse 141
Basel 4057
Switzerland

Dear Dr. Terranova,

Thank you for submitting your Research Article entitled "Drug-induced chromatin accessibility changes associate with sensitivity to liver tumor promotion". It is a pleasure to let you know that your manuscript is now accepted for publication in Life Science Alliance. Congratulations on this interesting work.

DISTRIBUTION OF MATERIALS:

Again, congratulations on a very nice paper. I hope you found the review process to be constructive and are pleased with how the manuscript was handled editorially. We look forward to future exciting submissions from your lab.

Sincerely,
